# Experiment design and bacterial abundance control extracellular H₂O₂ concentrations during 4 series of mesocosm experiments.

Mark J. Hopwood[1], Nicolas Sanchez[2], Despo Polyviou[3], Øystein Leiknes[2], Julian Gallego-Urrea[4], Eric P. Achterberg[1], Murat V. Ardelan[2], Javier Aristegui[5], Lennart Bach[1], Sengul Besiktepe[6], Yohann Heriot[1], Ioanna Kalantzi[7], Tuba Terbıyık Kurt[8], Ioulia Santi[7], Tatiana M. Tsagaraki[9], David Turner[4]

*Correspondence to:* Mark J. Hopwood (mhopwood@geomar.de)

1 GEOMAR Helmholtz Centre for Ocean Research Kiel, Germany
2 Norwegian University of Science and Technology, Trondheim, Norway
3 Ocean and Earth Science, National Oceanography Centre Southampton, United Kingdom
4 Marine Sciences, University of Gothenburg, Sweden
5 Instituto de Oceanografía y Cambio Global, IOCAG, Universidad de Las Palmas de Gran Canaria, ULPGC, Las Palmas, Spain
6 The Institute of Marine Sciences and Technology, Dokuz Eylul University, Turkey
7 Institute of Oceanography, Hellenic Centre for Marine Research, Heraklion, Greece
8 Department of Marine Biology, Faculty of Fisheries, Çukurova University, Turkey
9 Department of Biological Sciences, University of Bergen, Norway

## Abstract

The extracellular concentration of H₂O₂ in surface aquatic environments is controlled by a balance between photochemical production and the microbial synthesis of catalase and peroxidase enzymes to remove H₂O₂ from solution. In any kind of incubation experiment, the formation rates and equilibrium concentrations of ROS may be sensitive to both the experiment design, particularly to the regulation of incident light, and the abundance of different microbial groups, as both cellular H₂O₂ production and catalase/peroxidase enzyme production rates differ between species. Whilst there are extensive measurements of photochemical H₂O₂ formation rates and the distribution of H₂O₂ in the marine environment, it is poorly constrained how different microbial groups affect extracellular H₂O₂ concentrations, how comparable extracellular H₂O₂ concentrations within large scale incubation experiments are to those observed in the surface-mixed layer, and to what extent a miss-match with environmentally relevant concentrations of ROS in incubations could influence biological processes differently to what would be observed in nature. Here we show that both experiment design and bacterial abundance consistently exert control on extracellular H₂O₂ concentrations across a range of incubation experiments in diverse marine environments.

During 4 large scale (>1000 L) mesocosm experiments (in Gran Canaria, the Mediterranean, Patagonia and Svalbard) most experimental factors appeared to exert only minor, or no, direct effect on H₂O₂ concentrations. For example, in 3 of 4 experiments where pH was manipulated to 0.4-0.5 below ambient pH no significant change was evident in extracellular

$H_2O_2$ concentrations relative to controls. An influence was sometimes inferred from zooplankton density, but not consistently between different incubation experiments and no change in $H_2O_2$ was evident in controlled experiments using different densities of the copepod *Calanus finmarchichus* grazing on the diatom *Skeletonema costatum* (<1% change in [$H_2O_2$] comparing copepod densities from 1-10 $L^{-1}$). Instead, the changes in $H_2O_2$ concentration contrasting high/low zooplankton incubations appeared to arise from the resulting changes in bacterial activity. The correlation between bacterial abundance and extracellular $H_2O_2$ was stronger in some incubations than others ($R^2$ range 0.09 to 0.55), yet high bacterial densities were consistently associated with low $H_2O_2$. Nonetheless, the main control on $H_2O_2$ concentrations during incubation experiments relative to those in ambient, unenclosed waters was the regulation of incident light. In an open (lidless) mesocosm experiment in Gran Canaria, $H_2O_2$ was persistently elevated (2-6 fold) above ambient concentrations; whereas using closed high density polyethylene mesocosms in Crete, Svalbard and Patagonia $H_2O_2$ within incubations was always reduced (median 10-90%) relative to ambient waters.

## 1.0 Introduction

Reactive oxygen species (ROS), such as $H_2O_2$, are ubiquitous in surface aquatic environments due to photochemical formation (Van Baalen and Marler, 1966; Moore et al., 1993; Miller and Kester, 1994). Quantum yields for $H_2O_2$ formation increase with declining wavelength and so the ultraviolet (UV) portion of natural sunlight is a major source of $H_2O_2$ in surface aquatic environments (Cooper et al., 1988, 1994). Sunlight normalized $H_2O_2$ production rates therefore peak between wavelengths of 310-340 nm (Kieber et al., 2014). $H_2O_2$ is present at concentrations on the order of 10-100 nM in the ocean's surface mixed layer with its concentration generally declining sharply with depth (Price et al., 1998; Yuan and Shiller, 2001; Gerringa et al., 2004). Because its decay rate is slow (observed half-lives in seawater range from 10 to 120 h, Petasne and Zika 1997) compared to less stable ROS such as superoxide ($O_2^{-}$) and the hydroxyl radical ($OH^{-}$), extracellular $H_2O_2$ concentrations in surface waters show a pseudo-sinuous diurnal cycle, with elevated $H_2O_2$ concentrations occurring during daylight hours (Price et al., 1998).

$H_2O_2$ features as a reactive intermediate in the natural biogeochemical cycling of many compound groups including halocarbons (Hughes and Sun, 2016), trace metals (Moffett and Zika, 1987; Voelker and Sulzberger, 1996; Hansel et al., 2015) and dissolved organic matter (DOM) (Cooper et al., 1988; Scully et al., 2003). Previous work has highlighted the susceptibility of a broad range of marine biota to elevated extracellular $H_2O_2$ concentrations (Bogosian et al., 2000; Morris et al., 2011) and argued that measurable negative effects on metabolism occur in some marine species at $H_2O_2$ concentrations within the range of ambient surface-mixed layer concentrations (Morris et al., 2011; Baltar et al., 2013). Peroxidase and catalase enzymes are widely produced by marine microbes to lower extracellular $H_2O_2$ concentrations and these enzymes are the dominant sink for $H_2O_2$ in the surface marine environment (Moffett and Zafiriou, 1990; Angel et al., 1999). The reliance of some species including strains of *Prochlorococcus*, which do not produce such enzymes, on other 'helper' organisms to

remove extracellular $H_2O_2$ underpins a theory of reductive evolution, 'the Black Queen Hypothesis' (BQH) (Morris et al., 2012). BQH infers that because the removal of extracellular $H_2O_2$ by any species is a communal benefit, there is an energetic benefit to be gained to an individual species by losing genes associated with extracellular $H_2O_2$ detoxification. Loss of these genes continues to be favourable to individual species until only a minority of community members poses the ability to

remove $H_2O_2$, and the benefit of further loss would be offset by the negative effects of increasing extracellular $H_2O_2$ concentrations (Morris et al., 2012).

It is already acknowledged that laboratory incubation studies using buffered growth media are often conducted at $H_2O_2$ concentrations 2-10× higher than those found in the surface ocean (Morris and Zinser, 2013). We have previously

hypothesized that the same may be generally true for meso-scale experiments (Hopwood et al., 2018b) because the relative stability of $H_2O_2$ means that the enclosure of water at the ocean's surface within mesocosms can lead to elevated $H_2O_2$ concentrations. Yet there are presently few examples in the literature of incubation experiments where ROS concentrations are measured and therefore it is unknown how changes to other stressors, or changes to experimental design, affect extracellular ROS concentrations. In order to assess whether ROS could be a significant artefact in incubation experiments;

and to investigate how extracellular $H_2O_2$ concentrations respond to changes in DOC, pH and grazing pressure; here we collate data on $H_2O_2$ from a series of small to large scale (20-8000 L) incubation experiments with varying geographical location (Table 1).

## 2.0 Methods

Our rationale for the investigation of $H_2O_2$ trends during these 20-8000 L scale mesocosm and microcosm experiments is

that the experiment matrixes for each experiment permitted the changing of 1, 2 or 3 key variables (DOC, zooplankton, pH) whilst maintaining others (e.g. salinity, temperature, light) in a constant state across the mesocosm/microcosm experiment. The relationships between $H_2O_2$ and other chemical/biological parameters are therefore potentially easier to investigate than in the ambient water column where mixing and the vertical/lateral trends in $H_2O_2$ concentrations must also be considered. Additionally, two of the experiment designs described herein (see Table 1) were repeated in 3 geographic locations

facilitating direct comparisons between the experiment results with only limited mitigating factors concerning method changes.

### 2.1 Mesocosm set up and sampling

Eight incubation experiments (Table 1) were constructed using coastal seawater which was either collected through pumping from small boats deployed offshore, or from the end of a floating jetty. Three of these incubations were outdoor mesocosm

experiments (MesoPat, MesoArc and MesoMed) conducted using the same basic setup (based on that used in earlier experiments described by Larsen et al., 2015). For these three mesocosms, 10 identical cubic high density polyethylene

(HDPE) 1000-1500 L tanks were filled ~95% with seawater which was passed through nylon mesh (size as per Table 1) to remove mesozooplankton. The 10 closed mesocosm tanks were then held in position with a randomized treatment configuration and incubated at ambient seawater temperature. For MesoPat and MesoArc the mesocosms were tethered to a

jetty. For MesoMed the mesocosms were held in a pool facility at the Hellenic Centre for Marine Research which was continuously flushed with seawater to maintain a constant temperature. An extra HDPE container (to which no additions were made) was also filled to provide an additional supply of un-manipulated seawater (without zooplankton, DOC, or nutrient additions) for calibration purposes and baseline measurements on day 0. During MesoMed, this surplus container was incubated alongside the mesocosms for the duration of the experiment without any further additions/manipulation. In all

cases, these HDPE containers likely strongly attenuated natural UV radiation, compared to ambient waters, which is expected to negatively affect photochemical formation of $H_2O_2$ (Cooper et al., 1988, 1994).

| Label (Project) | Location | Month / year | Duration / days | Manipulated drivers | Scale / L | Site | Design Fig. S1 | $H_2O_2$ data available |
|---|---|---|---|---|---|---|---|---|
| MesoPat (Ocean Certain) | Comau fjord, Patagonia | Nov 2014 | 11 | DOC, grazing | 1000 | In-situ | I | Diurnal cycle. Limited time series |
| MultiPat (Ocean Certain) | Comau fjord, Patagonia | Nov 2014 | 8 | DOC, grazing, pH | 20 | Temperature controlled room | II | Final [$H_2O_2$] |
| MicroPat (Ocean Certain) | Comau fjord, Patagonia | Nov 2014 | 11 | DOC, grazing | 20 | Temperature controlled room | III | Final [$H_2O_2$] |
| MesoArc (Ocean Certain) | Kongsfjorden, Svalbard | July 2015 | 12 | DOC, grazing | 1250 | In-situ | I | Diurnal cycle |
| MultiArc (Ocean Certain) | Kongsfjorden, Svalbard | July 2015 | 8 | DOC, grazing, pH | 20 | Temperature controlled room | II | Limited time series |
| MesoMed (Ocean Certain) | Hellenic Centre for Marine Research, Crete | May 2016 | 12 | DOC, grazing | 1500 | Outdoor temperature controlled pool | I | Diurnal cycle, $H_2O_2$ time series, decay rates, $H_2O_2$ spiked incubation |

| | | | | | | | | |
|---|---|---|---|---|---|---|---|---|
| MultiMed (Ocean Certain) | Hellenic Centre for Marine Research, Crete | May 2016 | 9 | DOC, grazing, pH | 20 | Temperature controlled room | II | Final [$H_2O_2$] |
| Gran Canaria (The Future Ocean) | Taliarte Harbour, Gran Canaria | Mar 2016 | 28 | $pCO_2$ | 8000 | In-situ | IV | Diurnal cycle, $H_2O_2$ time series, $H_2O_2$ spiked incubation |

**Table 1 Details of experiments where $H_2O_2$ data were collected. Data from 8 separate experiments are presented, including 4 outdoor mesocosm experiments and 4 indoor microcosm/multistressor experiments. 'DOC' dissolved organic carbon.**

| Experiment | PAT (Patagonia) | ARC (Svalbard, Arctic) | MED (Crete, Mediteranean) | Gran Canaria |
|---|---|---|---|---|
| **Mesocosm** | MesoPat | MesoArc | MesoMed | Gran Canaria |
| Containers | HDPE 1000 L | HDPE 1250 L | HDPE 1500 L | Polyurethane 8000 L |
| Lighting | Ambient | Ambient | Ambient reduced ~50% with net | Ambient |
| Zooplankton treatment | +30 copepods $L^{-1}$ | +5 copepods $L^{-1}$ | +4 copepods $L^{-1}$ | NA |
| Macronutrient addition | N added as $NO_3$ | N added as $NH_4$ | N added as 50/50 $NH_4/NO_3$ | N added as $NO_3$ |
| Macronutrient addition timing | Daily | Daily | Daily | Day 18 only |
| Macronutrients added (per addition) | 1.0 µM $NO_3$, 1.0 µM Si, 0.07 µM $PO_4$ | 1.12 µM $NH_4$, 1.2 µM Si, 0.07 µM $PO_4$ (11.4 µM Si added on day 1) | 48 nM $NO_3$, 48 nM $NH_4$, 6 nM $PO_4$ | 3.1 µM $NO_3$, 1.5 µM Si, 0.2 µM $PO_4$ |
| Screening of initial seawater | NA | 200 µm | 140 µm | 3 mm |
| **Multistressor** | MultiPat | MultiArc | MultiMed | |
| Containers | HDPE collapsible 20 L | HDPE collapsible 20 L | HDPE collapsible 20 L | |
| Lighting | 36 W lamps | 36 W lamps | 36 W lamps | |
| Light regime | 15 h light / 9 h dark | 24 h light | 15 h light / 9 h dark | |
| Zooplankton treatment | +30 copepods $L^{-1}$ | +5 copepods $L^{-1}$ | +4 copepods $L^{-1}$ | |
| Macronutrient addition | Same as MesoPat | Same as MesoArc | Same as MesoMed | |
| Macronutrient addition timing | Daily | Daily | Daily | |
| Macronutrients added (per addition) | 1.0 µM $NO_3$, 1.0 µM Si, 0.07 µM $PO_4$ | 1.12 µM $NH_4$, 1.2 µM Si, 0.07 µM $PO_4$ | 48 nM $NO_3$, 48 nM $NH_4$, 6 nM $PO_4$ | |
| pH post adjustment | 7.54±0.09 | 7.76±0.03 | 7.64±0.02 | |
| pH pre-adjustment | 7.91±0.01 | 8.27±0.18 | 8.08±0.02 | |
| Screening of initial seawater | 200 µm | 200 µm | 140 µm | |
| Temperature / ℃ | 13-18 | 4.0-7.0 | 19.9-21.5 | |
| **Microcosm** | MicroPat | | | |
| Containers | HDPE collapsible 20 L | | | |
| Lighting | 36 W lamps | | | |
| Light regime | 15 h light / 9 h dark | | | |
| Containers | HDPE collapsible 20 L | | | |
| Grazing treatment | +30 copepods $L^{-1}$ | | | |
| Macronutrient addition timing | Daily | | | |

| Macronutrient addition | N was added as $NO_3$ |
|---|---|
| Macronutrients added (per addition) | 1.0 µM $NO_3$, 1.0 µM Si, 0.07 µM $PO_4$ |
| Screening of initial seawater | 200 µm |
| Temperature / ℃ | 14-17 |

**Table 2 Experiment details for each experiment. For a visual representation of experiment designs, the reader is referred to Supplementary Material. 'HDPE' high density polyethylene. 'NA' not applicable.**

The 10-mesocosm experiment design matrix was the same for MesoPat, MesoArc and MesoMed (Fig. S1, design I). For these 3 mesocosm experiments, zooplankton were collected one day in advance of requirement using horizontal tows at ~30 m depth with a mesh net equipped with a non-filtering cod end. Collected zooplankton were then stored overnight in 100 L containers and non-viable individuals removed by siphoning prior to making zooplankton additions to the mesocosm containers. After filling the mesocosms, zooplankton (quantities as per Table 2) were then added to 5 of the containers to create contrasting high/low grazing conditions. Macronutrients ($NO_3$/$NH_4$, $PO_4$ and Si) were added to mesocosms daily (Table 2). Across both the 5-high and 5-low grazing tank treatments, a dissolved organic carbon (DOC) gradient was created by addition of glucose to provide carbon at 0, 0.5, 1, 2 and 3 times the Redfield Ratio (Redfield, 1934) with respect to added $PO_4$. Mesocosm water was sampled through silicon tubing (permanently fixed into each mesocosm lid) immediately after mixing of the containers using plastic paddles (also mounted within the mesocosms through the lids) with the first 2 L discarded in order to flush the sample tubing.

A 4[th] outdoor mesocosm experiment (Gran Canaria) used 8 cylindrical polyurethane bags with a depth of approximately 3 m, a starting volume of ~8000 L and no lid or screen on top (Hopwood et al., 2018b). After filling with coastal seawater the bags were allowed to stand for 4 days. A pH gradient across the 8 tanks was then induced (on day 0) by the addition of varying volumes of filtered, $pCO_2$ saturated seawater (resulting in $pCO_2$ concentrations from 400-1450 µatm, treatments outlined Fig. S1 IV) using a custom-made distribution device (Riebesell et al., 2013). A single macronutrient addition (3.1 µM nitrate, 1.5 µM silicic acid and 0.2 µM phosphate) was made on day 18 (Table 2).

**2.2 Microcosm and multistressor set up and sampling**

A 10-treatment microcosm (MicroPat) incubation mirroring the MesoPat 10 tank mesocosm (treatment design as per Fig. S1 I, but with 6 × 20 L containers per treatment -one for each time point- rather than a single HDPE tank) and three 16-treatment multistressor experiments (MultiPat, MultiArc and MultiMed Fig. S1 II) were conducted using artificial lighting in temperature controlled rooms (Table 1, Fig. S1). For all 3 multistressor incubations (MultiPat, MultiArc and MultiMed) and the single microcosm incubation (MicroPat), coastal seawater (filtered through nylon mesh) was used to fill 20 L HDPE collapsible containers. The 20 L containers were arranged on custom made racks with light provided by a network of 36 W

lamps (Phillips, MASTER TL-D 90 De Luxe 36W/965 tubes). The number and orientation of lamps was adjusted to produce a light intensity of 80 µmol quanta $m^{-2} s^{-1}$. A diurnal light regime representing spring/summer light conditions at each fieldsite was used and the tanks were agitated daily and after any additions (e.g. glucose, acid or macronutrient solutions) in order to ensure a homogeneous distribution of dissolved components. Whilst the light condition for these experiments was selected to approximate the light intensity of ambient near-surface waters, the synthetic lighting is deficient in UV relative to

ambient sunlight-especially at low latitudes. In all 20 L scale experiments, macronutrients were added daily (as per Table 2). One 20 L container from each treatment set was 'harvested' for sample water each sampling day.

The experiment matrix used for the MicroPat incubation duplicated the MesoPat experiment design (Table 2) and thereby consisted of 10 treatments. The experiment matrix for the 3 multistressor experiments (MultiPat, MultiArc and MultiMed

outlined in Fig. S1 II) duplicated the corresponding mesocosm experiments at the same fieldsites (MesoPat, MesoArc and MesoMed), with one less C/glucose treatment and an additional pH manipulation (Table 2). The multistressor experiments thereby consisted of 16 treatments. pH manipulation was induced by adding a spike of HCl (trace metal grade) on day 0 only. For trace metal and $H_2O_2$ analysis, sample water from 20 L collapsible containers was extracted using a plastic syringe and silicon tubing which was mounted through the lid of each collapsible container.


Throughout, where changes in any incubation experiment are plotted against time, 'day 0' is defined as the day the experimental gradient (zooplankton, DOC, $pCO_2$) was imposed. Time prior to day 0 was intentionally introduced during some experiments to allow water to equilibrate with ambient physical conditions after container filling. $H_2O_2$ concentration varies on diurnal timescales and thus during each experiment where a time series of $H_2O_2$ concentration was measured,

sample collection and analysis occurred at the same time daily ($\pm 0.5$ h) and the order of sample collection was random.

### 2.3 Ancillary experiments

Four side experiments (1-4 below) were conducted to investigate potential links between bacterial/zooplankton abundance and extracellular $H_2O_2$ concentrations. Where specified, $H_2O_2$ concentrations were manipulated to form high, medium and low $H_2O_2$ conditions by adding aliquots of either a 1 mM $H_2O_2$ solution (prepared weekly from $H_2O_2$ stock) to increase $H_2O_2$

concentration, or bovine catalase (prepared immediately before use) to decrease $H_2O_2$ concentration. All treatments were triplicated. Catalase is photo-deactivated and biological activity to remove extracellular $H_2O_2$ follows the diurnal cycle (Angel et al., 1999; Morris et al., 2016), so catalase/$H_2O_2$ additions were conducted at sunset in order to minimize the additions required. Bovine catalase was used as received (Sigma Aldrich) with stock solutions prepared from frozen enzyme (stored at -20°C). De-natured catalase was prepared by heating enzyme solution to >90°C for 10 min. Plasticware used to

handle catalase solution was discarded. No adverse effects of measuring $H_2O_2$ in catalase-manipulated solutions were found. As $H_2O_2$ measurements were made >12 h after catalase addition, this may reflect catalase de-activation under the incubated conditions.

(1) In Gran Canaria a 5 day experiment was conducted, using 5 L polypropylene bottles. After filling with offshore seawater, and the addition of macronutrients which matched the concentrations added to the Gran Canaria mesocosm (3.1 µM nitrate, 1.5 µM silicic acid and 0.2 µM phosphate), bottles were incubated under ambient light and temperature conditions within Taliarte Harbor. (2) In Crete, a similar 7 day incubation was conducted in the HCMR pool facility using 20 L HDPE containers. Seawater was extracted from the baseline MesoMed mesocosm (no DOC or zooplankton addition) on day 11 and then incubated without further additions except for $H_2O_2$ manipulation. After day 5 no further $H_2O_2$ manipulations were made. (3) As per (2), seawater was withdrawn from the baseline MesoMed mesocosm on day 11 and then incubated without further addition except for $H_2O_2$ manipulation in 500 mL trace metal clean LDPE bottles under the artificial lighting conditions used for the MultiMed incubation. (4) A short term (20 h) experiment was conducted in trace metal clean 4 L HDPE collapsible containers to investigate the immediate effect of grazing on $H_2O_2$ concentrations. Filtered (0.2 µm, Satorius) coastal seawater (S 32.8, pH 7.9) water was stored in the dark for 3 days before use. The diatom *Skeletonema costatum* (NIVA-BAC 36 strain culture (CAA) from the Norsk Institutt for vannforskning (NIVA)) was used as a model phytoplankton grown in standard f/2 medium (Guillard and Ryther, 1962). Each treatment consisted of a total volume of 2 L seawater and contained macronutrients, 7.5 ml of the original medium (resulting in an initial chlorophyll a concentration of 3 µg $L^{-1}$ in the incubations) and treated seawater containing the copepod *Calanus finmarchichus* corresponding to each desired density. The light regime was produced with fluorescent lighting with a mean luminous intensity of 80-90 µmol $m^{-2}$ $s^{-1}$ and the temperature maintained at 10.5-10.9°C.

Light levels during all experiments (Table 1) were quantified using a planar Li-cor Q29891 sensor connected to a Li-cor Li-1400 data logger. Diurnal experiments measuring $H_2O_2$ concentrations in mesocosms or ambient surface (10 cm depth) seawater were conducted using flow injection apparatus with a continuous flow of seawater into the instrument through a PTFE line as described previously (Hopwood et al., 2018b). For extensive datasets, the diurnal range of $H_2O_2$ concentrations was determined as the difference between the means of the highest and lowest 10% of datapoints. Fe(II) and photochemically generated radical species can interfere with the luminol based chemiluminescence used to determine $H_2O_2$. In batch measurements, the >15 minute time delay between sample collection analysis (during which time the same in not exposed to light) is likely sufficient to minimize interference from these species which have much shorter half-lives and lower concentrations than $H_2O_2$ (Yuan and Shiller, 1999). For in situ continuous measurements a sample loop was intentionally introduced such that seawater was displaced from ambient light for 2 minutes prior to analysis. Some residual Fe(II) may have therefore remained leading to over-estimation of $H_2O_2$ concentrations. As Fe(II) concentrations were quantified in all of the experiments and ambient waters described herein (Hopwood et al., 2018a), we can determine that the maximum possible over-estimate of $H_2O_2$ concentrations for ambient waters during continuous analysis (diurnal timeseries) was <10%.

## 2.4 Chemical analysis

### $H_2O_2$

$H_2O_2$ samples were collected in opaque HDPE 125 mL bottles (Nalgene) which were pre-cleaned (1 day soak in detergent, 1 week soak in 1 M HCl, 3 rinses with de-ionized water) and dried under a laminar flow hood prior to use. Bottles were rinsed once with sample water, filled with no headspace and always analysed within 2 h of collection via flow injection analysis (FIA) using the Co(II) catalysed oxidation of luminol (Yuan and Shiller, 1999). FIA systems were assembled and operated exactly as per Hopwood et al., (2017) producing a detection limit of < 1 nM. Calibrations were run daily and with every new reagent batch using 6 standard additions of $H_2O_2$ (TraceSelect, Fluka) within the range 10-300 nM to aged (stored at room temperature in the dark for >48 h) seawater (unfiltered).

### Macronutrients

Dissolved macronutrient concentrations (nitrate+nitrite, phosphate, silicic acid; filtered at 0.45 µm upon collection) were measured spectrophotometrically the same day as sample collection (Hansen and Koroleff, 2007). For experiments in Crete (MesoMed, MultiMed), phosphate concentrations were determined using the 'MAGIC' method (Rimmelin and Moutin, 2005). The detection limits for macronutrients thereby inevitably varied slightly between the different mesocosm/microcosm/multistressor experiments (Table 1), however this does not adversely affect the discussion of results herein.

### Carbonate chemistry

$pH_T$ (except where stated otherwise, 'pH' refers to the total pH scale reported at 25ºC) was measured during the Gran Canaria mesocosm using the spectrophotometric technique of Clayton and Byrne (1993) with m-cresol purple in an automated Sensorlab SP101-SM system using a 25ºC-thermostatted 1 cm flow-cell exactly as per González-Dávila et al., (2016). pH during the MesoPat/MicroPat/MultiPat experiments was measured similarly as per Gran Canaria using m-cresol. During MesoArc/MultiArc/MesoMed/MultiMed experiments pH was measured spectrophotometrically as per Reggiani et al., (2016).

### Biological parameters

Chlorophyll a was measured by fluorometry as per Welschmeyer (1994). Bacterial production was determined by incorporation of tritium-labelled leucine ($^3$H-Leu) using the centrifugation procedure of Smith and Azam (1992). Conversion of leucine to carbon (C) was done with the theoretical factor 3.1 kg C mol$^{-1}$ leucine. In Gran Canaria, flow cytometry was conducted on 2 mL water samples which were fixed with 1% paraformaldehyde (final concentration), flash frozen in liquid $N_2$ and stored at -80ºC until analysis. Samples were analysed (FACSCalibur, Becton Dickinson) with a 15 mW laser set to excite at 488 nm (Gasol and del Giorgio, 2000). Subsamples (400 µL) for the determination of heterotrophic bacteria were stained with the fluorochrome SybrGreen-I (4 µL) at room temperature for 20 min and run at a flow rate of 16 µL min$^{-1}$. Cells were enumerated in a bivariate plot of 90° light scatter and green fluorescence. Molecular Probes latex beads (1 µm) were used as internal standards. In Crete (MesoMed/MultiMed), the flow cytometry was conducted similarly except for the

following minor changes: samples were fixed with 0.5% glutaraldehyde (final concentration), yellow-green microspheres (1 and 10 μm diameter, respectively) were used as internal references during the analysis of bacterial and nanoflagellate populations, and the flow rate was 79-82 μL min$^{-1}$. Subsamples (7-50 L) for zooplankton composition and abundance were preserved in 4% borax buffered formaldehyde solution and analysed microscopically.

## 3.0 Results

### 3.1 $H_2O_2$ time series during outdoor mesocosm incubations; MesoMed and Gran Canaria

In order to understand the controls on $H_2O_2$ concentrations in incubations, time series of $H_2O_2$ are first presented for those experiments with the highest resolution data. Also of interest are trends in bacterial productivity following the observation that $H_2O_2$ decay constants appear to correlate with bacterial abundance in a range of natural waters (Cooper et al., 1994). The concentration of $H_2O_2$ was followed in all treatments on all sampling days during the Gran Canaria and MesoMed mesocosms. In Gran Canaria, comparing mean (±SD) $H_2O_2$ in all mesocosms across a $pCO_2$ gradient (400-1450 μatm) with $H_2O_2$ in ambient seawater outside the mesocosms, $H_2O_2$ was generally elevated within the mesocosms compared to ambient seawater (ANOVA $p < 0.05$ for all treatments compared to ambient conditions). The mean and median ambient $H_2O_2$ concentration throughout the experiment was at least 40% lower than that in any mesocosm treatment (Fig. 1). This included the 400 μatm mesocosm which received no additions of any kind until the nutrient spike on day 18. The only exception was a short time period under post-bloom conditions when bacterial abundance peaked and daily integrated light intensity was relatively low (compared to the mean over the duration of the experiment) for 3 consecutive days (experiment days 25-27, Hopwood et al., 2018). No clear trend was observed with respect to the temporal trend in $H_2O_2$ and the $pCO_2$ gradient. $H_2O_2$ concentration in the baseline $pCO_2$ treatment was close to the mean (400-1450 μatm) for the duration of the 28 day experiment.

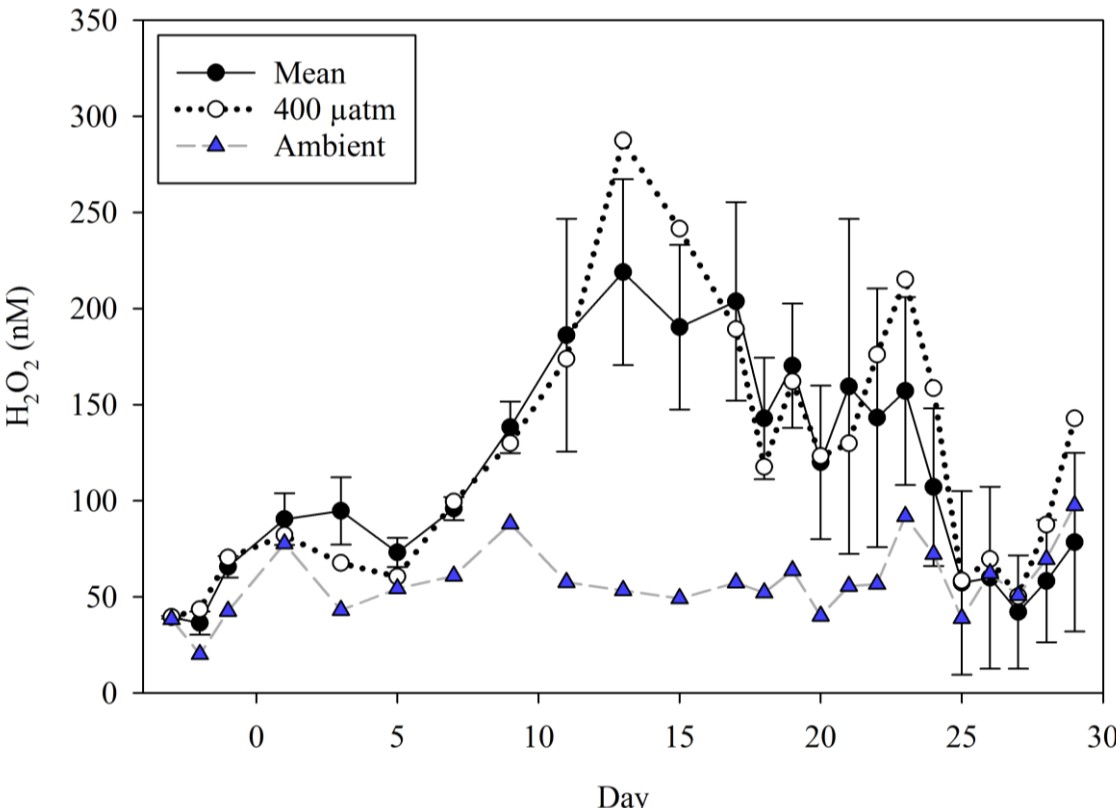

**Figure 1: A summary of H$_2$O$_2$ over the duration of a pCO$_2$ gradient mesocosm in Gran Canaria. Data from Hopwood et al., (2018). The mean (± SD) H$_2$O$_2$ from all pCO$_2$ treatments is contrasted with the concentration in ambient surface seawater immediately outside the mesocosms. In addition to its inclusion in the mean, the baseline 400 µatm pCO$_2$ treatment is shown separately to allow comparison with ambient surface seawater.**

During MesoMed (Fig. 2) an additional mesocosm tank was filled (Tank 11) and maintained without any additions (no

macronutrients, no DOC, no zooplankton) alongside the 10 mesocosm containers. As per the Gran Canaria mesocosm, H$_2$O$_2$

concentrations were also followed in ambient seawater throughout the duration of the MesoMed experiment. MesoMed was

however conducted in an outdoor pool facility, so the ambient concentration of H$_2$O$_2$ in coastal seawater refers to a site

approximately 500 m away from the incubation pool. Ambient H$_2$O$_2$ was generally higher than that observed within the

mesocosm with a median concentration of 120 nM around midday (Fig. 2(a)).

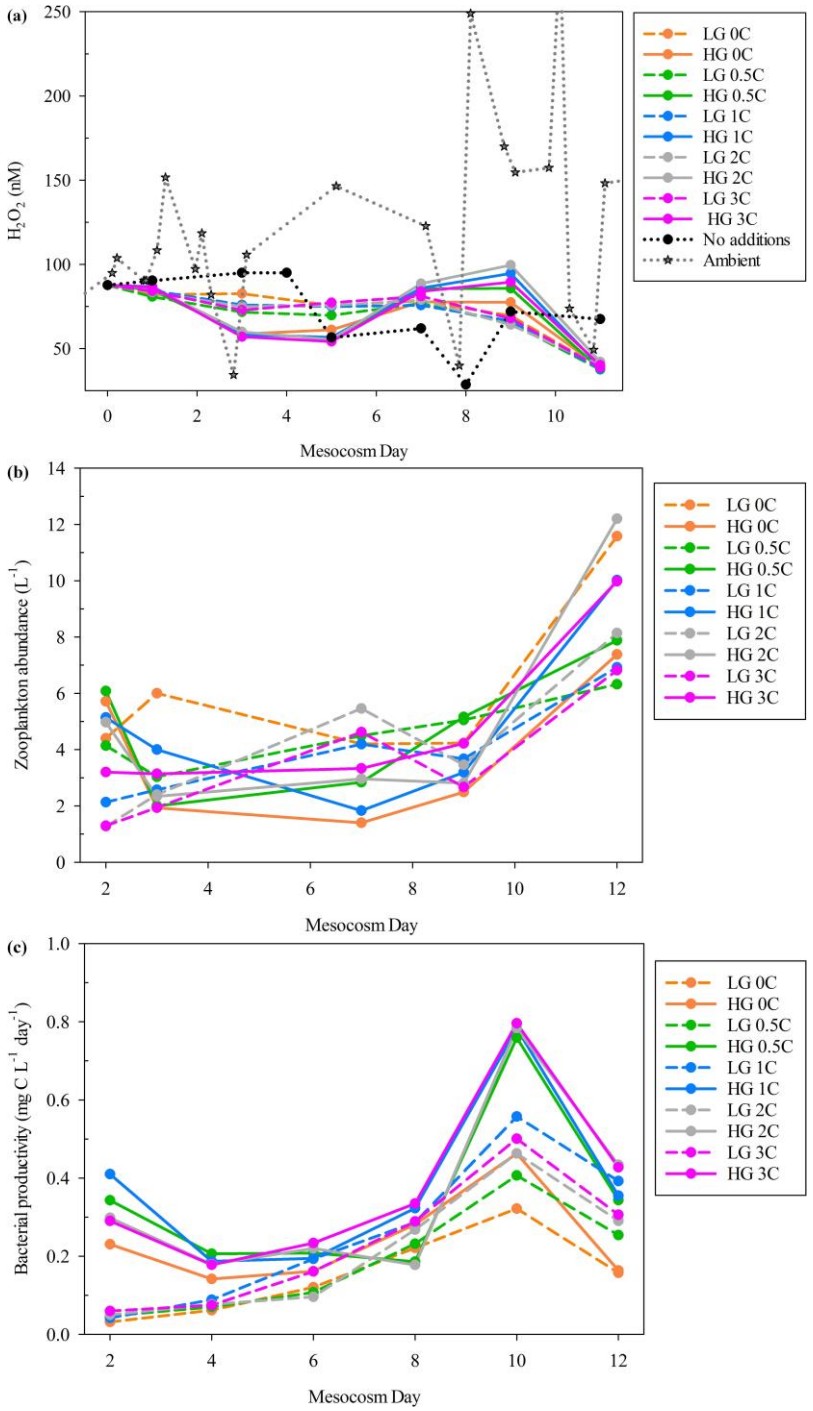


**Figure 2: (a) $H_2O_2$ in all mesocosms during MesoMed in Gouves, Crete. A 10-treatment matrix (as per Fig. S1) was used (b) Zooplankton abundances showed a rapid convergence in the HG/LG (high grazing/low grazing) status of the mesocosms after day 2 (c) The trend in bacterial productivity showed broad similarity within the HG and LG treatment groups. 'C' refers to the ratio of added carbon (glucose) in each treatment.**

$H_2O_2$ during the MesoMed experiment was relatively constant in terms of the range of concentrations measured over the 11 day duration of the experiment (Fig. 2), especially when compared to the Gran Canaria mesocosm (Fig. 1). A notable clustering of the high ('HG') and low ('LG') zooplankton tanks was clearly observed between days 1 and 9 (Fig. 2) (addition of zooplankton took place immediately after day 1 sampling). $H_2O_2$ concentration in the high zooplankton tanks initially declined more strongly than the low zooplankton tanks, then re-bounded together after day 5 (Fig. 2). Dilution experiments

to estimate zooplankton grazing and zooplankton abundance (Fig. 2) both suggested that between days 3 and 7, the high/low grazing status of the mesocosms converged i.e. grazing declined in the tanks to which zooplankton had initially been added and increased in the tanks to which no zooplankton had been added such that initial 'high/low' grazing labels became obsolete (Rundt, 2016). $H_2O_2$ concentration declined sharply in all treatments on day 11, except in the no-nutrient addition mesocosm, coinciding with a pronounced increase in zooplankton abundance and occurring just after bacterial productivity

peaked in all treatments (Fig. 2). $H_2O_2$ decay rate constants in the dark (measured using freshly collected seawater at the MesoMed fieldsite over 24 h and assumed to be first order) were 0.049 $h^{-1}$ (unfiltered) and 0.036 $h^{-1}$ (filtered, Satorius 0.2 µm) corresponding to half-lives of 14 h and 19 h, respectively, which are within the range expected for coastal seawater (Petasne and Zika, 1997).

**3.2   $H_2O_2$ trends during 20 L scale indoor MultiPat, MultiMed and MicroPat incubations**

A sustained decline in $H_2O_2$ concentration was found whenever ambient seawater was moved into controlled temperature rooms with artificial diel light cycles (e.g Fig. 3) which were used to incubate all 20 L scale multistressor and microcosm experiments discussed herein (Table 1). Final $H_2O_2$ concentrations in these 20 L scale experiments were thereby generally low compared to those measured in corresponding ambient surface waters and to the corresponding outdoor experiments in the same locations with natural lighting (and higher UV light exposure).

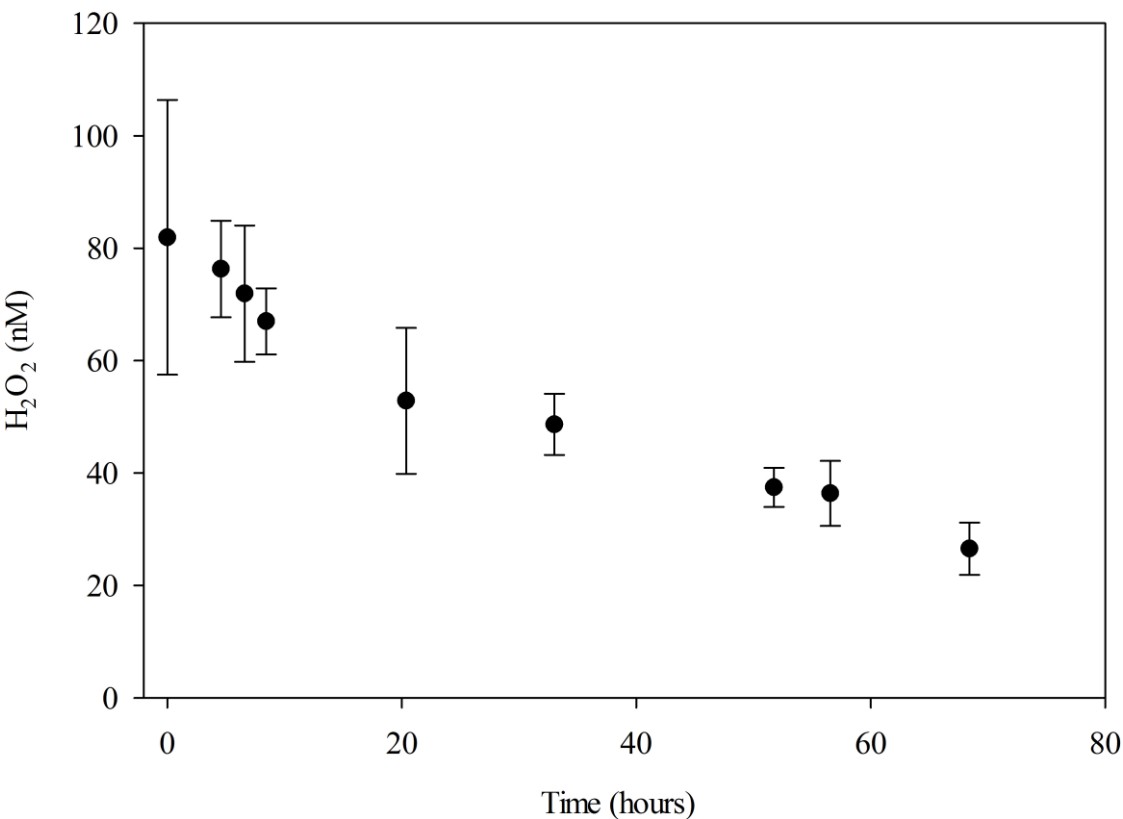


**Figure 3: Seawater from MesoMed (without macronutrient, DOC or zooplankton amendment) was used to fill a 20 L HDPE container which was then incubated under the synthetic lighting used in the MultiMed experiment for 72 h with regular sub-sampling for analysis of $H_2O_2$.**

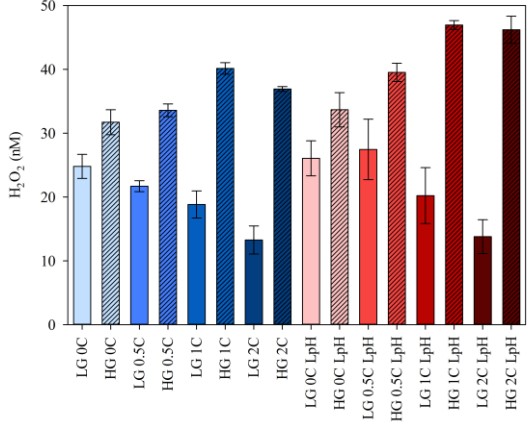

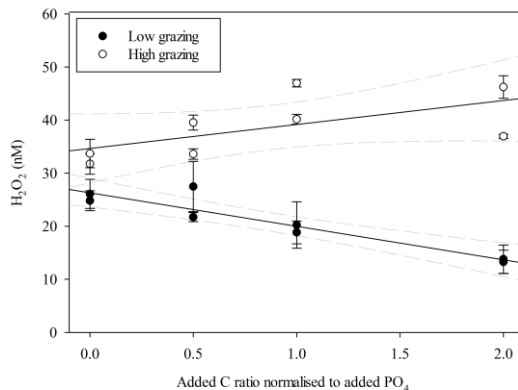

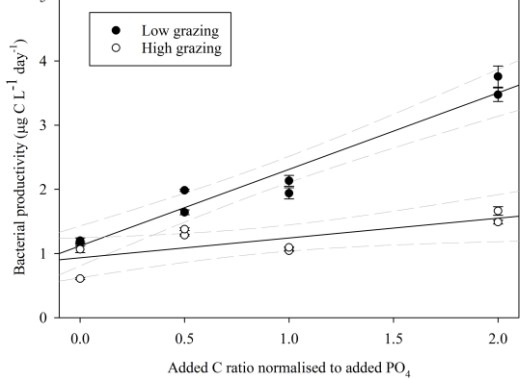

**Figure 4: (a) $H_2O_2$ concentrations at the end of the MultiMed experiment (Day 9). Ambient pH (blue), low pH (red); high grazing (hashed); carbon (C, glucose) added at 0, 0.5, 1.0, 1.5 and 2.0 × Redfield carbon: phosphate ratio. (b) Plotting both ambient and low pH datapoints together, which exhibited no statistically significant difference in $H_2O_2$ concentrations, final $H_2O_2$ concentration showed contrasting trends between high and low grazing treatments over the added C gradient. 95% confidence intervals are shown. (c) Bacterial productivity, measured via leucine incorporation, during the same experiments.**

H$_2$O$_2$ concentrations by the end of the MultiMed experiments (day 9) were universally low compared to the range found in comparable ambient waters and the outdoor mesocosm incubation conducted at the same fieldsite (Fig. 2). As was the case in the MesoMed experiment, a clear difference was noted between H$_2$O$_2$ concentrations in the high and low zooplankton addition treatments (Fig. 4 (b)), with the high grazing always resulting in higher H$_2$O$_2$ concentrations (t test, p <0.001). Any effect of pH was less obvious, with similar results obtained between ambient (initially 8.08 ± 0.02) and low (initially 7.64 ± 0.02) pH treatments (Fig. 4 (a)) and thus low and ambient pH treatments are not distinguished in Fig. 4 (b) and (c). An effect of the imposed C gradient on H$_2$O$_2$ concentrations was notable in both the high and low grazing treatments, yet the effect operated in the opposite direction (Fig. 4 (b)). In high grazing treatments, increasing C corresponded to increasing extracellular H$_2$O$_2$ concentrations (linear regression coefficient 4.5 ± 2.3); whereas in low grazing treatments, increasing C corresponded to decreasing extracellular H$_2$O$_2$ concentrations (linear regression coefficient -6.3 ± 0.97). Bacterial productivity increased with added C in both high (linear regression coefficient 0.31 ± 0.1) and low grazing treatments (linear regression coefficient 1.2 ± 0.1), but there was a more pronounced increase under low grazing conditions (Fig. 4 (c)).

At the end of the MultiPat experiment (day 8), H$_2$O$_2$ concentrations were similarly low compared to ambient surface waters at the Patagonia fieldsite (Fig. 5 (a)), although there was a greater range of results. In the low pH treatment (initially 7.54 ± 0.09), H$_2$O$_2$ concentrations were significantly higher (Mann-Whitney Rank Sum test p=0.02) compared to the unmodified pH treatment (initially 8.01 ± 0.02). However, two of the low pH treatments with particularly high H$_2$O$_2$ were outliers (defined as 1.5 IQR) when considering the data as consisting of two pH groups. Without these two datapoints, there would be no significant difference between H$_2$O$_2$ in high and low treatments (p=0.39). Contrary to the results from the MultiMed experiment (Fig. 4), there was no significant difference between high/low grazing treatments (Mann-Whitney Rank Sum test p=0.65). Bacterial productivity also showed similar results between the high and low grazing treatments (Fig. 5 (b)). Data from day 5 (the last day bacterial productivity was measured) showed a similar gradient in increased bacterial productivity with added C for both high/low grazing treatment groups (linear regressions HG 0.64, R$^2$ 0.70 and LG 0.72, R$^2$ 0.92).

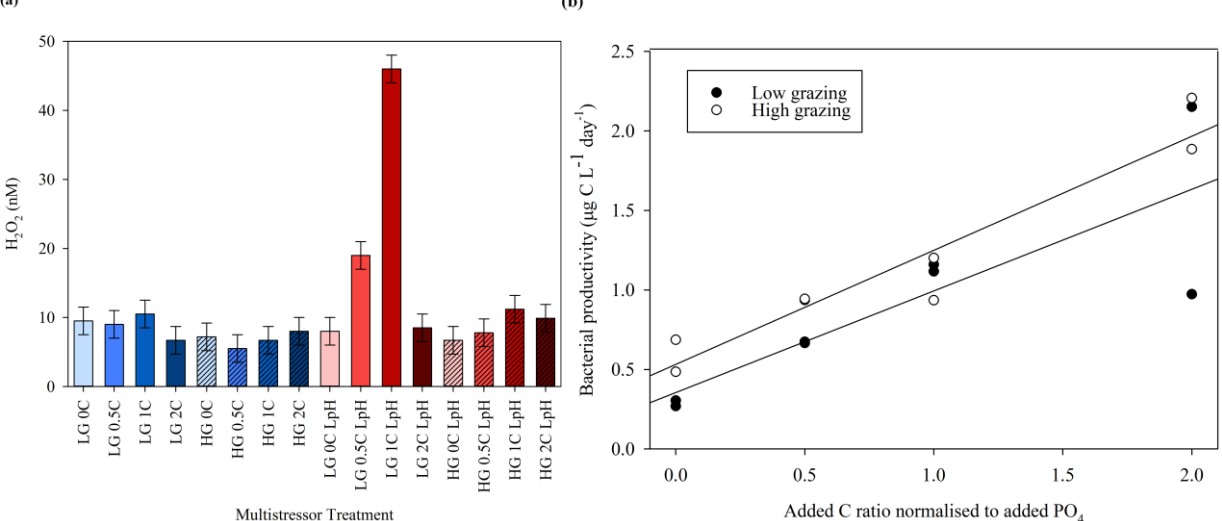

**Figure 5: (a) H₂O₂ concentrations at the end of the MultiPat experiment. Normal pH (blue), low pH (red); high grazing (hashed); DOC added at 0, 0.5, 1.0, and 2.0 × Redfield carbon (C):phosphate ratio indicated by increasing colour density. (b) Plotting both high and low pH datapoints together (which exhibited no statistically significant difference in H₂O₂ concentrations), bacterial productivity showed similar trends between the HG and LG treatments.**

The MicroPat experiment, also conducted using 20 L HDPE containers and artificial lighting, yielded no clear trend with respect to $H_2O_2$ concentrations over the imposed C gradient (Fig. 6, day 11), but the high grazing treatments were associated with higher $H_2O_2$ concentrations (t-test, p=0.017). Bacterial productivity was not systematically different across the high/low grazing treatment groups, nor was there as clear a trend in bacterial productivity with respect to the added C gradient (Fig. 6 (c)) compared to the MultiPat (Fig. 5 (b)) or MultiMed (Fig. 4 (c)) experiments.

(a)

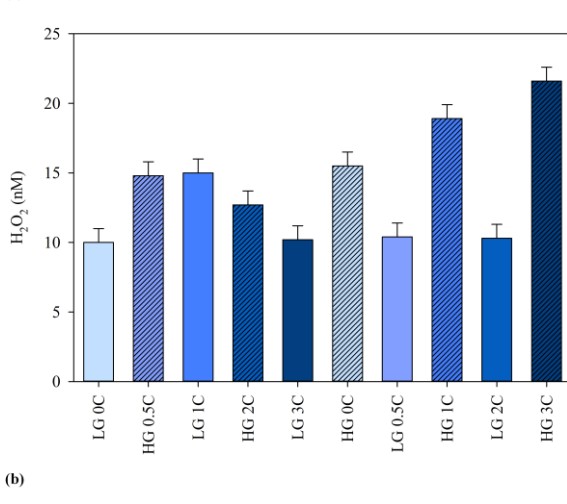

(b)

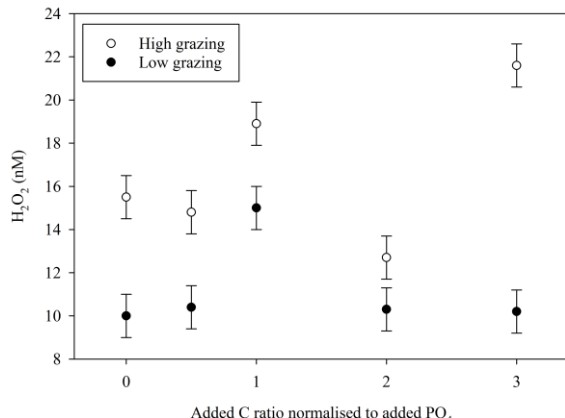

(c)

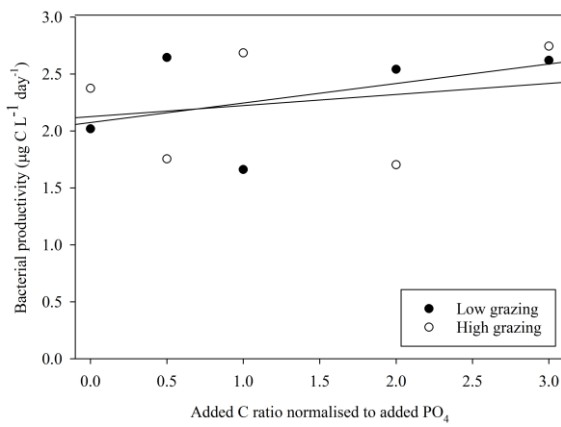

**Figure 6: (a) H₂O₂ concentrations at the end of the MicroPat experiment. High grazing treatments are hashed; DOC added at 0, 0.5, 1.0, 2.0 and 3.0 × Redfield carbon (C):phosphate ratio indicated by increasing colour density. (b) No clear trend was evident across the DOC gradient, but high grazing was consistently associated with higher H₂O₂ concentration. (c) Bacterial productivity in the same experiment.**


### 3.2 Diel cycling of $H_2O_2$; results from the Mediterranean

In addition to the trends observed over the duration of multi-day incubation experiments, a diurnal variability in $H_2O_2$
concentrations is expected. The diurnal cycle of $H_2O_2$ concentrations during MesoMed was followed in the no-addition tank (number 11) over 2 days with markedly different $H_2O_2$ concentrations (Fig. 4). An additional cycle was monitored at a nearby coastal pier (Gouves) for comparative purposes. The mean difference between mid-afternoon and early-morning $H_2O_2$ could also be deduced from discrete time points collected over the experimental duration in seawater close to the pool facility. All time series are plotted against local time (UTC+1). Sunrise/sunset was as follows: (May 15) 06:15, 20:17; (May
19) 06:12, 20:20. All three time series showed the expected peak in $H_2O_2$ concentrations during daylight hours, but the timing of peak $H_2O_2$ concentration and the range of concentrations observed differed between mesocosms and coastal seawater. The intraday range in $H_2O_2$ concentrations in Gouves, and the afternoon peak in $H_2O_2$, (Fig. 7) was similar to that observed previously in Gran Canaria (Hopwood et al., 2018b). Yet both the mesocosm diurnal time series exhibited notably limited diurnal ranges and peak $H_2O_2$ concentration occurred earlier, around midday (Fig. 7), than in coastal waters.

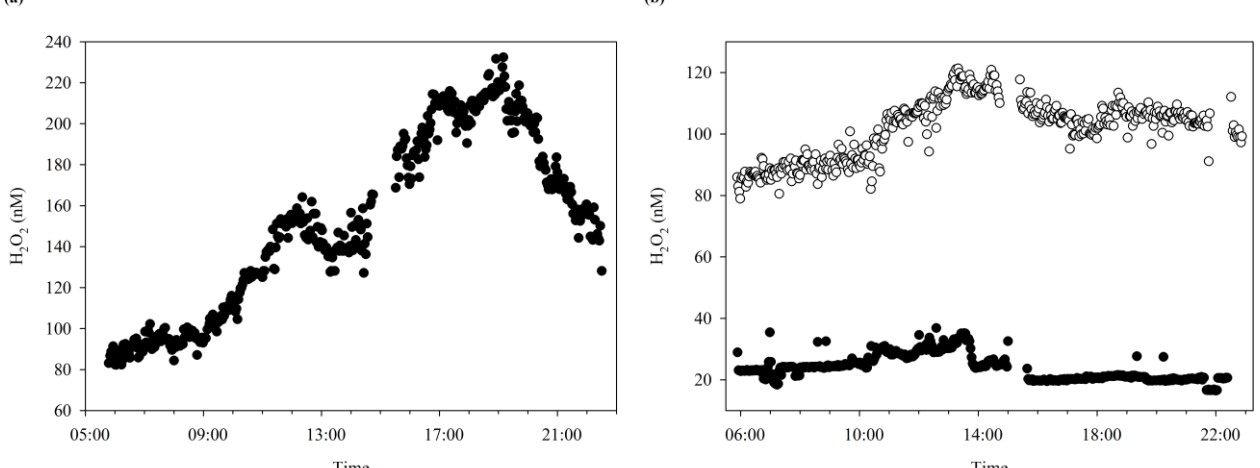


**Figure 7: (a) Diurnal cycling of $H_2O_2$ in coastal seawater (Gouves, Crete 17 May) and (b) in the no addition tank (number 11) during the MesoMed mesocosm on May 15 (open circles) and May 19 (closed circles) 2016 (experiment days 4 and 8, respectively).**

### 3.3 Ancillary experiments to investigate links between microbial groups (bacterial, zooplankton) and extracellular $H_2O_2$

In addition to comparing $H_2O_2$ concentrations in different incubation experiments to assess the effect of experiment setup on extracellular $H_2O_2$ concentrations, potential links between microbial groups and $H_2O_2$ were explored. The MesoPat/Arc/Med, MicroPat and MultiPat/Arc/Med experiments all included a high/low zooplankton addition treatment (Table 2). Over a 20 h incubation (4 h darkness, 16 h light) in an experiment with varying concentrations of copepods (0-25 $L^{-1}$) grazing on an intermediate density of a diatom (initially 3 µg $L^{-1}$ chlorophyll a), $H_2O_2$ concentrations showed no inter-treatment differences
(Fig. 8). A diatom was selected as phytoplankton stock because cell normalized $H_2O_2$ production rates for diatoms appear to be generally at the low end of the observed range for phytoplankton groups (Schneider et al., 2016). Fe(II) concentration

(measured at the same time as per Hopwood et al., 2018a) also appeared to be unaffected by the copepod density as the difference between treatments was almost negligible (<0.04 nM).

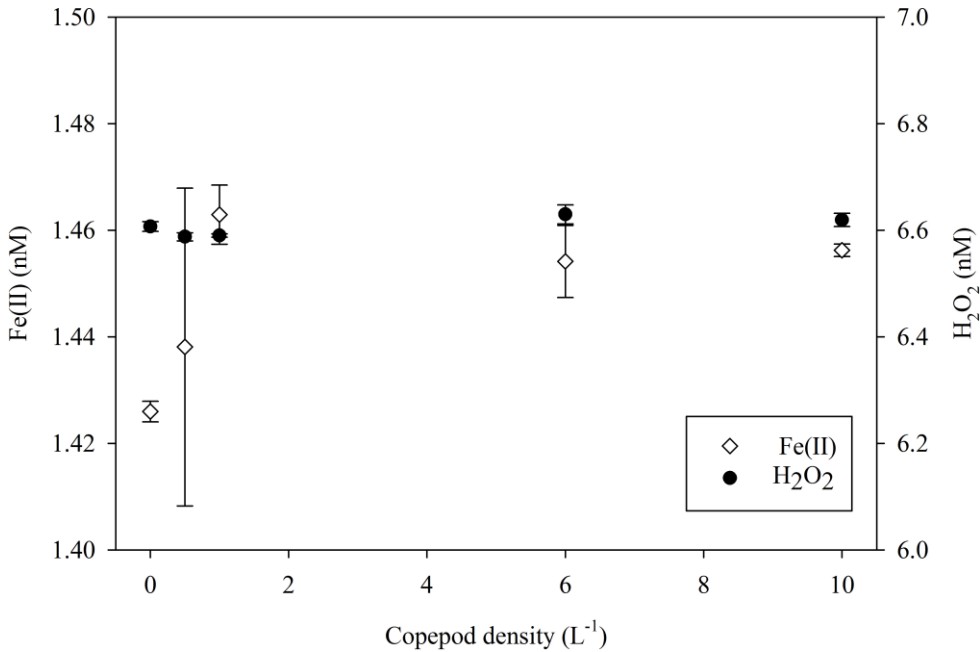

**Figure 8: $H_2O_2$ and Fe(II) concentrations in a culture of diatoms growing in coastal seawater after 20 h of incubation with a zooplankton gradient imposed by addition of copepods.**

At the end of the MesoMed experiment, seawater (extracted from the baseline treatment from the mesocosm on day 11) was used in two side experiments. During both the extracellular $H_2O_2$ concentration was manipulated, with each treatment triplicated. In all cases the mean (±SD) of three replicate treatments is reported. The high-medium-low $H_2O_2$ concentration gradient used in each experiment was determined by considering the ambient concentration of $H_2O_2$ in the mesocosms (e.g. Fig. 2) and in ambient seawater close to the mesocosm facility. After the first daily $H_2O_2$ measurements were made, the required spikes to maintain the desired $H_2O_2$ gradient were calculated based on measured rates of $H_2O_2$ decay. $H_2O_2$ and catalase spikes were then added at sunset followed by gentle mixing.

A test specifically to investigate the effect of the multistressor/microcosm experimental set up on bacterial activity was conducted in 500 mL trace metal clean LDPE bottles under the artificial lighting conditions (~80 µmol quanta m$^{-2}$ s$^{-2}$) used for the MultiMed experiment. $H_2O_2$ concentrations again verified that manipulation with $H_2O_2$ spikes successfully created a low, medium and high $H_2O_2$ treatment (mean for triplicate low/medium/high treatments: $40 \pm 2$, $120 \pm 6$, $230 \pm 7$ nM $H_2O_2$). Bacterial production showed no statistically significant (ANOVA, p=0.562) difference between triplicate low ($1.69 \pm 0.28$ µg C L$^{-1}$ day$^{-1}$), medium ($1.30 \pm 0.60$ µg C L$^{-1}$ day$^{-1}$) and high ($1.29 \pm 0.56$ µg C L$^{-1}$ day$^{-1}$) $H_2O_2$ treatments.

For a concurrent manipulation in the Mediterranean using 20 L HDPE containers incubated outdoors, a gradient in $H_2O_2$ concentrations was similarly imposed. These manipulations successfully produced a clear gradient of $H_2O_2$ conditions with relatively consistent $H_2O_2$ concentrations within each triplicated set (Fig. 9 (a)). After day 5 no further manipulations were conducted and $H_2O_2$ accordingly began to converge towards the medium (no $H_2O_2$ spike, no active catalase spike) treatment. Flow cytometry, conducted on low/medium/high samples at $8 \times 24$ h intervals over the experiment duration, measured no significant (ANOVA, $p > 0.05$) difference between the 3 treatments for cell counts of any group (bacteria are shown as an example, Fig. 9 (b)).

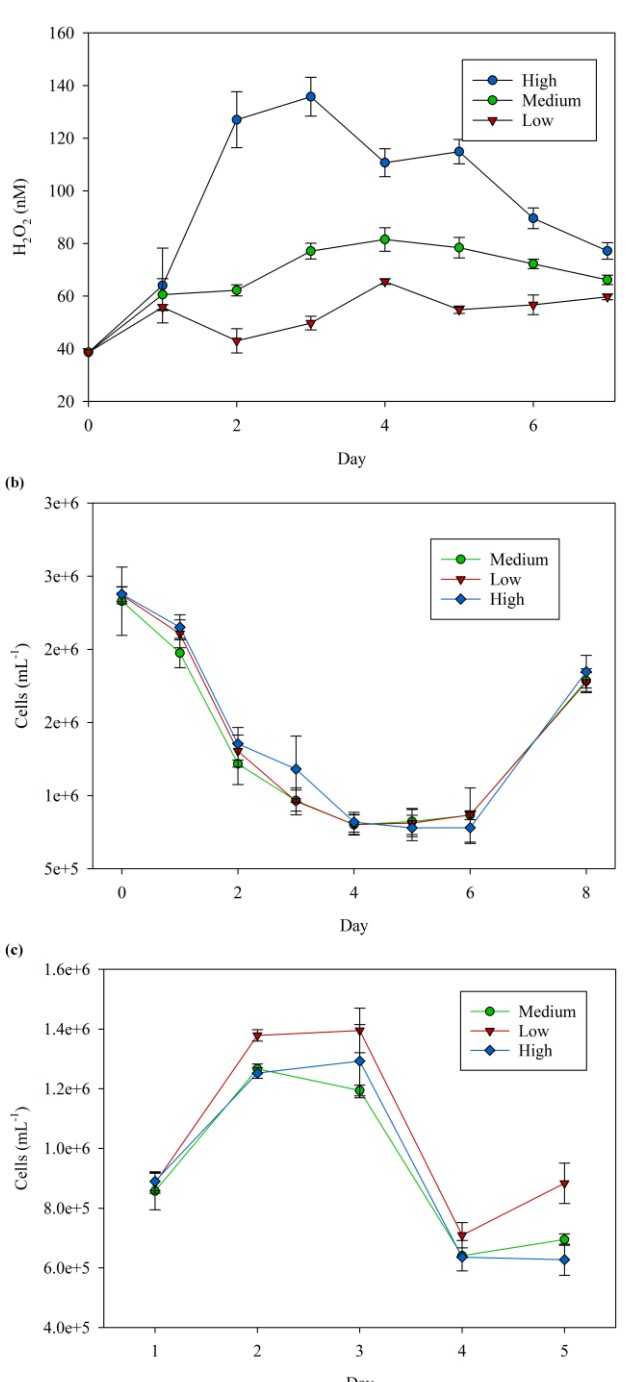

Figure 9: (a) $H_2O_2$ gradient during the 20 L scale Mediterranean side experiment where a $H_2O_2$ gradient was created with $H_2O_2$ spikes and catalase (b) bacteria abundance during the same Mediterranean experiment (c) bacteria abundance for a similar incubation in Gran Canaria. Mean and standard deviations of triplicate treatments are plotted in all cases.

395

A similar side experiment was conducted in Gran Canaria, but one critical difference was the addition of macronutrients at the start of the experiment, as per the mesocosm at the same location (Table 2). Measurement of $H_2O_2$ concentrations, which were initially $43 \pm 1$ nM (mean of all $3 \times 3$ replicates at day 0), confirmed that a gradient was maintained over the 5-day duration of the experiment (mean $210 \pm 113$, $62 \pm 14$ and $47 \pm 8$ nM in the high, medium and low $H_2O_2$ treatments, respectively). Some modest shifts in phytoplankton group abundance were observed over the duration of this experiment. Slightly higher cell counts of bacteria were consistently observed in the low $H_2O_2$ treatment relative to the medium and high $H_2O_2$ treatment (Fig. 9 (c)). Only the difference between the low and medium/high treatments was significant (ANOVA, $p=0.028$)- no significant difference was found between the medium and high $H_2O_2$ treatments (ANOVA, $p=0.81$).

## 4    Discussion

### 4.1 Bacteria, zooplankton and extracellular $H_2O_2$ trends

During all meso/multi/micro experiments and the Gran Canaria mesocosm (Table 1), data was available on the abundance of bacteria and zooplankton throughout the experiment. We focus on zooplankton because of the top-down control they may exert on primary production and the potential for grazing to release trace species into solution which may affect $H_2O_2$ biogeochemistry. Bacteria were a key focus because of the hypothesis that bacteria are, via the production of peroxidase/catalase enzymes, the main sink for $H_2O_2$ in surface aquatic environments (Cooper et al., 1994).

Throughout, no clear effect was evident of changing pH on $H_2O_2$ concentrations. The 440-1450 µatm $pCO_2$ gradient applied in Gran Canaria, which corresponded to a pH range of approximately 7.5-8.1, and the contrasting ambient/low pH (a reduction in pH of 0.4-0.5 from ambient waters was imposed) applied during 3 multistressor incubations (Table 2) exhibited no obvious change in equilibrium extracellular $H_2O_2$ concentration. Similarly no change was evident in Gran Canaria when contrasting the diurnal cycling of $H_2O_2$ in the 400 and 1450 µatm $pCO_2$ treatments (Hopwood et al., 2018b). In the incubation experiments, whenever there was a sustained difference in extracellular $H_2O_2$ concentrations between treatment groups (MesoMed Fig. 2 and MultiMed Fig. 4), the main difference arose between 'high' and 'low' zooplankton addition treatments. However, determining the underlying reason for this was complicated by the shifts in zooplankton abundance during the experiments (e.g. Fig. 2 (b)).

The MultiPat (Fig. 5) and MicroPat (Fig. 6) incubations showed no significant effect of increased zooplankton abundance on extracellular $H_2O_2$. Two reasons for this can be considered. First, in Patagonia the initial ratio of zooplankton between the high and low treatments was the smallest of the experiments herein (17:14) and thus a large difference might not have been anticipated compared to the experiments where this initial ratio was always considerably higher. However, the mean ratio of HG:LG zooplankton by the end of MultiPat had increased to 9:5. By comparison, during MesoMed (when the HG:LG

zooplankton abundance converged during the experiment, Fig. 2(b)) the HG:LG ratio after day 1 varied within the range 0.32-1.6 and thus the final ratio of 1.8 in MultiPat was not particularly low. A more distinct difference however arose in bacterial productivity (Fig. 5 (b)). Unlike MesoMed, MultiPat and MicroPat showed little difference in bacterial productivity between the high and low grazing treatments. Thus the effects of zooplankton with respect to shifts in the abundance of other microbial groups (rather than grazing itself) may be the underlying reason why extracellular $H_2O_2$ concentrations sometimes,

but not consistently, changed between high and low grazing treatments. Second, in any case $H_2O_2$ concentrations at the end of the Patagonian experiments (MesoPat, MicroPat and MultiPat) were also very low (almost universally <20 nM) and thus the signal:noise ratio unfavourable for detecting differences between treatments.

     Furthermore, the effect of higher zooplankton populations was not a consistent positive/negative change in extracellular

$H_2O_2$. During the post-nutrient addition phase in Gran Canaria, the single treatment with slower nutrient drawdown (mesocosm 7) due to high grazing pressure exhibited relatively high $H_2O_2$ (Hopwood et al., 2018b). During MesoMed, increases in zooplankton abundance coincided with decreases in $H_2O_2$ concentration (Fig. 2). Similarly, during MultiMed (Fig. 4), the effect of adding zooplankton was the same; high zooplankton treatments exhibited low $H_2O_2$ concentration. As high zooplankton are correlated during some experiments, and anti-correlated in others, with $H_2O_2$, the underlying cause did

not appear to be that $H_2O_2$ is generally produced by the process of grazing (i.e. as a by-product of feeding). Further support for this argument was found in the results of a simple side experiment adding copepods (*Calanus finmarchichus*) to a diatom culture (*Skeletonema costatum*) (Fig. 8). No measurable change in extracellular $H_2O_2$ concentration was found at higher densities of copepods either during a 16 h light incubation, or after 4 h of incubation in the dark (Fig. 8). There are two obvious limitations in this experiment; a different result may have been obtained with a different combination of copepod

and phytoplankton, and standard f/2 medium contains the ligand ethylenediaminetetraacetic acid (EDTA) which may affect $H_2O_2$ formation rates by complexing trace species involved in $H_2O_2$ cycling (e.g. dissolved Fe and Cu). Nonetheless, it is known that cellular ROS production rates vary at the species level (Schneider et al., 2016; Cho et al., 2017), so shifts in species composition as a result of zooplankton addition are a plausible underlying cause of changes in extracellular $H_2O_2$ concentration. We summarise that any correlation between $H_2O_2$ and zooplankton thereby appears to have arisen from the

resulting change in the abundance of microbial species, and thus the net contribution of biota to extracellular $H_2O_2$ concentration, rather than from the act of grazing itself.

     Bacteria are expected to be a dominant $H_2O_2$ sink in most aquatic environments (Cooper et al., 1994). Here the correlation between extracellular $H_2O_2$ and bacteria cell counts was much stronger in some experiments than others ($R^2$ from 0.09-0.55).

A key reason for this may simply be the generally low $H_2O_2$ concentrations measured in most of our experiments. At the low $H_2O_2$ concentrations of <50 nM observed during most experiments, the influence of any parameter on $H_2O_2$ removal would be more challenging to determine from an analytical perspective due to reduced signal:noise ratio. However, the $H_2O_2$-defence mechanism of organisms may also be sensitive to ambient $H_2O_2$ concentrations. Morris et al., (2016) suggest that

microbial communities exposed to high $H_2O_2$ have elevated $H_2O_2$ defences. If the microbial communities here exhibited a dynamic response to $H_2O_2$ concentrations in terms of their extracellular $H_2O_2$ removal rates, this would dampen the correlation between bacterial abundance and $H_2O_2$ concentrations. Combing all available $H_2O_2$ concentrations for which the corresponding total bacterial cell counts are available (Fig. 10) from all experiments (except the side experiments where $H_2O_2$ was manipulated using catalase or $H_2O_2$ spikes), provides some limited evidence for the dominance of bacteria as a $H_2O_2$ sink. There was a notable absence of high-$H_2O_2$, high-bacteria datapoints in any experiment (Fig. 10). The observed distribution is therefore consistent with a scenario where bacteria dominate $H_2O_2$ removal, but other factors (possibly including experiment design, see s4.2) can also lead to low $H_2O_2$ conditions independently of bacterial abundance.

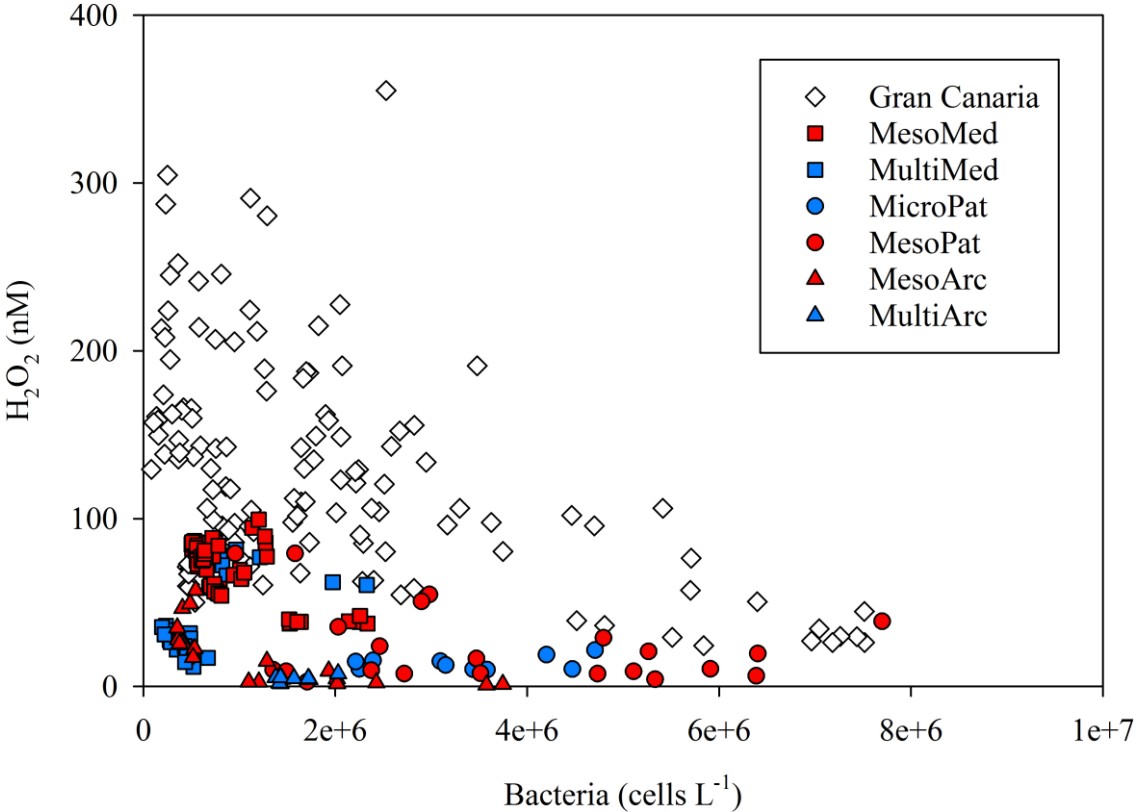

**Figure 10: Bacterial cell counts and $H_2O_2$ for all available data from all incubation experiment time-points where both measurements were made within 24 h of each other.**

## 4.2 Changes in extracellular $H_2O_2$ due to experiment design

When all available $H_2O_2$ datapoints were normalized to ambient $H_2O_2$ at the respective fieldsite, which varied between our locations (Table 3), some qualitative inter-experiment trends were evident. Experiments incubated with artificial lighting (MultiPat/Arc/Med and MicroPat) generally exhibited the lowest concentrations, while higher normalized $H_2O_2$ concentrations were observed in the closed HDPE mesocosms (MesoMed, MesoPat, MesoArc) and then the open Gran

Canaria mesocosm experiment (Fig. 11 (b) and (c)). This is not surprising considering the light arrangements for these experiments (Table 2). The Gran Canaria experiment was practically unshaded with surface seawater exposed to natural sunlight. The closed HDPE mesocosms (MesoMed, MesoPat, MesoArc) experienced natural sunlight but after attenuation through 1-2 cm of HDPE plastic. Whilst the transmission of different light wavelengths through these HDPE containers was not tested during our experiments, 1-2 cm of polyethylene should strongly attenuate the UV component of sunlight. The 20 L scale experiments (MultiMed, MultiPat, MultiArc and MicroPat) were conducted using identical synthetic lighting with lamps selected to as closely as possible replicate the wavelength distribution of natural sunlight. However, the fluorescent light distribution is still deficient, relative to sunlight, in wavelengths <400 nm, which is the main fraction of light that drives $H_2O_2$ formation in surface seawater (Kieber et al., 2014), and these containers still mitigated the limited UV exposure with a 1 mm HDPE layer which would further reduce any UV component of incoming light.

| Location | Season | Latitude | Salinity | Temperature / °C | $H_2O_2$ / nM |
|---|---|---|---|---|---|
| Taliarte, Gran Canaria | March 2016 | 30.0° N | 36.6-36.8 | 18-19 | 10-50 |
| Gouves, Crete | May 2016 | 35.3° N | NA | 19-20[a] | 34-410[b] |
| Comau fjord, Patagonia | November 2014 | 42.4° S | 3.9-12.8 | 9.7-13 | 120-680 |
| Kongsfjorden, Svalbard | July 2015 | 78.9° N | 9.0-35.2 | 5.0-9.0 | 10-100 |

**Table 3. Range of water properties in freshly collected coastal seawater at each site where the mesocosms were conducted. 'NA' not applicable. [a] Temperature of pool facility at HCMR, [b] Coastal seawater approximately 500 m from HCMR facility.**

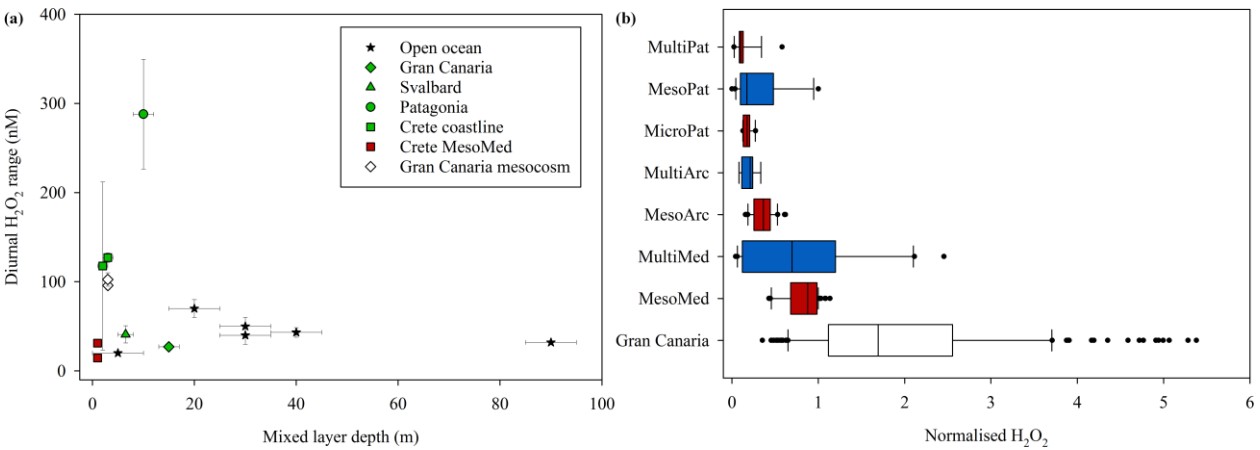

**Figure 11: (a) Observed diurnal ranges in $H_2O_2$ concentrations. Black stars show literature surface marine values and green shapes in-situ experiments corresponding to experiment field site locations (b) $H_2O_2$ across all experiments as a fraction of ambient $H_2O_2$. For the Meso/Multi fieldsites (Mediterranean, Arctic and Patagonia) red bars are outdoor mesocosms and blue shapes indoor incubations. Median, 10th/25th/75th/90th percentiles and all outliers are shown.**

During all periods when high resolution $H_2O_2$ time series were obtained, a clear diurnal trend was observed with a peak in $H_2O_2$ concentration occurring around midday (Fig. 7). Yet the range of concentrations within the two MesoMed diurnal

experiments ($31.2 \pm 2.3$ nM and $14.5 \pm 2.7$ nM) was limited compared to those observed previously within a Gran Canaria mesocosm ($96 \pm 4$ and $103 \pm 8$ nM, Hopwood et al., 2018). For comparison, the diurnal ranges reported in further offshore surface waters of the Atlantic, Gulf of Mexico and sub-tropical equatorial Pacific along the Peruvian shelf are 20-30 nM (Yuan and Shiller, 2001), 40-70 nM (Zika et al., 1985) and 40 nM,[1] respectively with no clear systematic trend associated with changes in mixed layer depth (Fig. 11 (a)). Within mesocosms and the coastal mesocosm fieldsites, the range was more

variable. Notably, the MesoMed diurnal ranges (15 and 31 nM) were considerably lower than that observed at two corresponding coastal sites (one monitored over a single diurnal cycle, $127 \pm 5$ nM; one at regular intervals over the duration of the experiment, $118 \pm 94$ nM). Whereas, conversely, for the Gran Canaria mesocosm the ~100 nM diurnal range was much greater than that observed ($27.0 \pm 3.1$ nM) in ambient surface waters (Fig. 11 (a)).

There are inevitably limits to what can be determined from contrasting available data on $H_2O_2$ concentration from multiple incubation experiments due to the different experiment designs (see Table 2). Yet the experiment setup with respect to moderating light during an experiment appears to be critical to establishing the equilibrium $H_2O_2$ concentration and can either enhance or retard the extracellular concentration of $H_2O_2$ during the experiment. The diurnal range plotted for all mesocosm experiments reflected increased $H_2O_2$ concentrations during daylight hours. This concentration range was

suppressed in the closed HDPE containers (e.g. MesoMed), yet enhanced in open polyurethane bags (Gran Canaria). Whilst not explicitly measured, a likely factor to explain this difference is the reduced UV light exposure to water within sealed HDPE containers compared to open polyurethane bags. During the multistressor and microcosm experiments, incubated indoors in 20 L HDPE containers, the diurnal range in $H_2O_2$ concentrations was suppressed sufficiently that no increase in $H_2O_2$ was apparent during simulated daylight hours. Lighting conditions, especially the UV component of light which acts as

a major source of photochemical $H_2O_2$ formation in natural waters, for the experiments therefore could explain both the contrasting change in the diurnal range of $H_2O_2$ (Fig. 11a), and the shift in the gradient between bacteria and $H_2O_2$ under different experiment conditions (Fig. 10).

### 4.3  ROS, bacteria and the Black Queen Hypothesis

Results from experiments where $H_2O_2$ concentrations were manipulated were mixed. In a side experiment after MesoMed,

there was no evidence of strong positive or negative effects of $H_2O_2$ concentrations on any specific microbial group (Fig. 9). In Gran Canaria, under different experimental conditions (macronutrients were added, whereas for the MesoMed side experiment no macronutrient spike was added), a small increase in bacterial abundance was found at low $H_2O_2$ concentrations (+27%, Fig. 9 (c))). This result alone should be interpreted with caution, as the addition of catalase can have other effects in addition to lowering $H_2O_2$ concentration (Morris, 2011), yet it is intriguing to consider the role of $H_2O_2$ as an

intermediate in the cycling of DOM alongside the role of bacteria as the dominant $H_2O_2$ sink.

---

[1] Unpublished data kindly provided by Insa Rapp (GEOMAR).

Photochemistry both enhances the lability of DOM (Bertilsson and Tranvik, 1998; Keiber et al., 1990) (thus making it more bioavailable as a substrate for bacteria) and causes the direct photochemical oxidation of DOM into dissolved inorganic carbon (Miller and Zepp, 1995; Granéli et al., 1996) (thus rendering it unavailable as a substrate for bacteria). ROS may enhance both of these processes, but few attempts have been made to determine the effect of manipulating ROS concentrations on photochemical DOM degradation rates, especially in the marine environment and at nanomolar concentrations (Pullin et al., 2004). Yet in experiments using furfuryl alcohol to suppress ROS in lake water, the rate of dissolved inorganic carbon formation when exposed to light decreased 20% and bacterial populations when later incubated in this ROS-quenched water were 4-fold higher than water with 'normal' ROS activity (Scully et al., 2003) implying that ROS removal was beneficial for bacteria. The results of experiments conducted in freshwater environments are not directly applicable to the marine environment, due to the different conditions in the ambient water column, but it is plausible that a similar mechanism underpinned the increase in bacteria abundance observed in Gran Canaria following the artificial lowering of $H_2O_2$ concentrations (Fig. 9). A large difference in bacterial populations between the presence and absence of some ROS species (Scully et al., 2003) raises interest in how important an influence changes in ROS concentration could be on the availability of DOM for bacterial productivity in the surface marine environment when more subtle changes are made to ambient $H_2O_2$ concentrations. If heterotrophic bacteria are the dominant $H_2O_2$ sink (Cooper et al., 1994), which the observed trend between bacterial abundance and extracellular $H_2O_2$ across a broad range of incubation experiments is consistent with (Fig. 10), this is also interesting in light of the Black Queen Hypothesis. BQH (Morris et al., 2012) assumes that the sole major benefit of producing enzymes that remove extracellular $H_2O_2$ is protection against the oxidative stress associated with high $H_2O_2$ concentrations- which is a communal benefit (Zinser, 2018). Yet, if increasing extracellular $H_2O_2$ concentrations accelerate the degradation of labile DOM to dissolved inorganic carbon, a second benefit of $H_2O_2$ removal in productive waters is the enhanced availability of this DOM to heterotrophs. Thus, under some circumstances, it could possibly be more favourable for heterotrophic species to maintain genes associated with the removal of $H_2O_2$ than autotrophic species because, in addition to the shared communal benefit of lowering oxidative stress, heterotrophs would suppositionally benefit more directly than autotrophs from the enhanced stability of labile DOM under low $H_2O_2$ conditions. However, whilst $H_2O_2$ is a reactive species, at the concentrations present in the marine environment the direct effects of changing $H_2O_2$ concentration on the abundances of different microbial groups (e.g. Fig. 9) are clearly minor. A specific challenge with determining the effect(s) of $H_2O_2$ concentration on any biogeochemical processes, and vice-versa, is that the diurnal variability in $H_2O_2$ concentration is always large compared to inter-treatment differences in $H_2O_2$ concentration within individual experiments (e.g. Fig. 11). High resolution data is therefore clearly required to properly interpret $H_2O_2$-microbial interactions and to better quantify the subtle links between $H_2O_2$ cycling and microbial functioning.

# 5    Conclusions

Extracellular $H_2O_2$ concentrations and bacterial abundances over a broad range of incubation experiments conducted in the marine environment support the hypothesis that bacterially produced enzymes are the dominant $H_2O_2$ sink. If heterotrophic bacteria are generally the main sink for $H_2O_2$ in surface marine environments, it is of interest to determine whether changes in extracellular $H_2O_2$ concentration measurably affect the photochemical transformation of DOM transformation to dissolved inorganic carbon. If increasing equilibrium ROS concentrations decreases the availability of labile DOM as a substrate for heterotrophs, this may affect which group/species produce catalase/peroxidase enzymes in productive waters.

It was apparent from comparing multiple experiments that incubation experiment design is also a strong influence on $H_2O_2$ concentrations. Closed HDPE mesocosms exhibited concentrations 10-90% lower than those expected in the corresponding ambient seawater, whereas an open (lidless) mesocosm exhibited concentrations 2-6 fold higher than ambient seawater. The diurnal range in $H_2O_2$ within incubations was also correspondingly increased in experiments where $H_2O_2$ concentration was artificially high, and vice-versa where $H_2O_2$ concentration was artificially low, suggesting enhanced, or reduced, photochemical stress over the diurnal cycle. Incubated experiments thus poorly mimic the biogeochemistry of reactive photo-chemically formed trace species.

## 4    Author Contributions

MH, DP, JG, EA, DT and MA designed the study. MH, NS, DP, ØL, JG, MA, JA, SB, YH, IK, TK and TT undertook work at one or more of the mesocosm/microcosm/multistressor experiments. MH, NS, DP, ØL, JG, JA, LB, SB, YH, TK, IS and TT conducted analytical work. MH, NS, DP, SB and TT interpreted the data. MH coordinated the writing of the manuscript with input from other authors.

## 5    Acknowledgements

The Ocean Certain and KOSMOS/PLOCAN teams assisting with all aspects of experiment logistics and organisation are thanked sincerely for their efforts. Labview software for operating the $H_2O_2$ FIA system was designed by P Croot, M Heller, C Neill and W King. Financial aid from the European Commission (OCEAN-CERTAIN, FP7- ENV-2013-6.1-1; no: 603773) is gratefully acknowledged. JA was supported by a Helmholtz International Fellow Award, 2015 (Helmholtz Association, Germany).

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
