# Peer review of "Experiment design and bacterial abundance control extracellular H₂O₂ concentrations during 4 series of mesocosm experiments."

_Biogeosciences, 2018_

## Short Comment (SC1) · 11 Jul 2018

This work provides large scale mesocosm experiments to elucidate how microbial groups affect extracellular H2O2 concentrations and other related questions. It has shown that the high bacterial densities were associated with low H2O2.

This manuscript generally reads well and presents a good rationale of research. However, the study could be significantly improved with the addition of missing details on the methodology used in experiment design, as well as statistical support.

The major issue is that there are so many variables in this work, which have not been fully considered regarding to the result interpretation. All these variables could play a great role in affecting the extracellular $H_2O_2$ concentration while the rationale to use these variables were not explained well and when the conclusion could not be obtained between microbial groups and $H_2O_2$ concentrations if all other variables were playing great role in it. These variables include (not limited to): zooplankton concentrations, different bacterial community, temperature, nutrient (concentrations and chemicals), light (light cycle and light intensity), DOC and pH. For example: In Glippa et al., 2018, "Vehmaa et al. [21] found that a 3 degrees rise in temperature increased the antioxidant capacity (ORAC, Oxygen Reactive Absorbance Capacity) in Acartia copepods by almost 15%, and they measured a 2-fold increase also in oxidative damage, measured as lipid peroxidation".

Specific comments: The line numbers started over on each page. It is better to have continuous line number from the beginning to the end of the manuscript. P9 L27: Is there statistics to support the "$H_2O_2$ was generally elevated"? P11 L9-L10: It is hard to get the conclusion of "this trend closely matched that observed in zooplankton biomass" by only eyeballing it, especially when the 5th day of zooplankton biomass was not shown in the figure. P12 L13: Statistics would be helpful to support "a clear difference was noted between". P13 L7-L8: Again statistics would be needed to the statement "there was a more pronounced increase". P13 L1-L13: Regarding to the statements, "In the low pH treatment (initially 7.54 $\pm$ 0.09), $H_2O_2$ concentrations were significantly higher (Mann-Whitney Rank Sum test p 0.02) compared to the unmodified pH treatment (initially 8.01 $\pm$ 0.02)". Only by eye-balling it, it showed the LG0.5C LpH and LG 1C LpH have higher concentration of $H_2O_2$. Is this statement based on only these two data points? Regarding to the statistics p value, it would be helpful if it is equal to, less than or greater than some certain number by indicating with corresponding symbols. P15 L8-L13: It would be great to put these discussions after (Table 1) under Discussion. P16 L16-L17: Regarding to this statement, "Bacterial production showed no statistically significant (ANOVA, P 0.562) difference between low, medium

and high H2O2 treatments.", there is no data to support it. Is it related with Fig. 9(c)? P17 L3: The author claimed there is NO significant difference while the p value is less than 0.05.

Figure 1: There is line to indicate the Mean H2O2. However, it is not clear on how to get this Mean. Figure 2: Is there any interpretation on the big variation of H2O2 in ambient? Is there replicates to have error bar? Statistics would be helpful here to show the difference between HG/LG status.

Figure 7: It would be great to show diurnal cycling of H2O2 in two continuous days.

---

## Referee Comment (RC1) · Anonymous Referee #1 · 24 Jul 2018

The goal of this study was to determine if aspects of an experimental design could inadvertently affect the photochemical or biological production of hydrogen peroxide (H2O2), thus altering the outcome of the study. This was tested by analyzing the compiled data from multiple coastal mesocosm experiments and determining which factors or aspects of the experimental design caused a change in H2O2 concentration compared to the ambient concentration found in surrounding seawater. Based upon their analysis, the authors concluded that the isolation of seawater within a mesocosm, alterations to light intensity, and changes to bacterial abundance were responsible for

variations in H2O2 concentration between the mesocosm vessels and the surrounding seawater. This study represents an interesting opportunity to observe how standard methods of experimental design (mesocosms) could potentially influence experimental outcomes in marine environments. Additionally, this study is unique in how the authors explore the effect of organisms of higher trophic levels upon H2O2 concentrations. The authors were able to provide convincing evidence supporting the importance of bacterial communities in modulating H2O2 concentrations in the ocean.

Major comments: A major conclusion of the paper is that light treatment (ambient versus artificial) has a big impact on the H2O2 concentrations in the mesocosm experiment. While this is supported by the figures, it is difficult to tell which light treatments are used for each figure, and there is no indication in Table 1 if the mesocosms are exposed to sunlight or light bulbs. Along these lines, there is essentially no discussion of the differences in light exposure, particularly the ability of UV in sunlight to generate the H2O2, and this should be mentioned in both the introduction and the discussion. The authors attempt to demonstrate how aspects of an experimental design (structure of vessel, setup, nutrient addition, increased stress) could affect the concentration of H2O2. While changes in H2O2 are measurable in all mesocosm experiments and are potentially attributable to a particular aspect of the experiment, the observed changes in H2O2 concentration are small with respect to total daily production of H2O2. All but one of the mesocosm experiments have H2O2 concentrations below 100nM and ranges of variation between 20-50nM. The prospect of changes in H2O2 concentration such as these recorded altering experimental outcome for microbial activity and DOC decay seems unlikely, without cited support. Pg. 18 lines 24-26 – As stated here, no clear trends can be defined between H2O2 concentration and grazer abundance when considering all datasets used. Perhaps it would be beneficial to focus more intently upon the aspect of bacterial abundance and its effect upon H2O2 concentrations instead? Along with above comment, bacterial abundance is an integral part of this study's conclusions yet only 2 figures give any data on how their abundances are changing. Inclusion of cells count data for the other experiments and datasets would

strengthen this major argument of the paper.

Minor comments:

The authors claim that the isolation of seawater in mesocosm vessels allows for the accumulation of H2O2. This is discussed throughout the manuscript but notably in Figure 1. on pg. 9 line 22-32 and pg. 21 line 1-11. In Figure 1, the authors claim that there is no clear trend between H2O2 and pCO2 concentration, leading them to conclude that changes in H2O2 are due to the enclosure used to house the water. Does this graph show H2O2 concentrations in unamended seawater within one of the polyurethane bags used, i.e. is the baseline 400atm a control? If not, then H2O2 production cannot solely be attributed to the container used. In Figure 1 is it possible that the microbes are nutrient depleted by day 8-9, and the increase in H2O2 is due to their decline in abundance? This would also explain why the H2O2 concentration decreases around day 18 when the nutrient addition was made. Axis labels throughout manuscript are misleading. H2O2 / nM should be shown as H2O2 (nM), etc. In Figure 2 panel a, the H2O2 concentrations for ambient seawater and LG 2C treatment are difficult to discern. Consider a different representation of the data. Pg. 20 lines 15-20 – The authors are comparing H2O2 production ranges from open ocean environments to those measured in coastal environments. In Table 2 on pg. 20, the upper H2O2 concentrations listed for the Crete and Patagonia locations are significantly higher than any data shown in previous figures from those same locations. Pg. 21 lines 13-14 – Were individual microbial groups ever quantified? Or was this observation made from total cell counts? Figures 4a and 5a: are these data from the same experiment? The values for "LG 1C" look different in these figures, as one example.

---

## Author Comment (AC1) · 20 Dec 2018

As Dr Ma kindly provided a comment and, later, a review; our responses to these comments are detailed under the review (below).
* * *
[Figure]

---

## Author Comment (AC2) · 20 Dec 2018

Two reviewers are thanked for insightful comments on the submitted text.

Please note that in addition to the comments by reviewers on this text, a companion manuscript concerning a different aspect of the same mesocosm experiments was also recently reviewed for this journal. As it is highly desirable to have a consistent use of terminology between these (and other in prep.) texts concerning the experiment set up, the following change has also been made to this text in order to maintain consis-

tency: The names of the major experiments has been standardized throughout the text and we have been careful to use only one specific term of reference for each experiment. The mesocosm/microcosm/mutlistressor experiments are now termed MesoPat/MesoArc/MesoMed/MultiPat/MultiArc/MultiMed/MicroPat/Gran Canaria.

(Previously the term 'MesoPat' was used to loosely refer to the field campaign which included a trio of mesocosm/multistressor/microcosm experiments, but this was found to be confusing, 'MesoPat' now refers exclusively to the 1000 L scale mesocosm experiment conducted in Patagonia, 'MultiPat' to the 20 L multistressor experiment at the same fieldsite location etc...).

Anonymous Referee #1 The goal of this study was to determine if aspects of an experimental design could inadvertently affect the photochemical or biological production of hydrogen peroxide (H2O2), thus altering the outcome of the study. This was tested by analyzing the compiled data from multiple coastal mesocosm experiments and determining which factors or aspects of the experimental design caused a change in H2O2 concentration compared to the ambient concentration found in surrounding seawater. Based upon their analysis, the authors concluded that the isolation of seawater within a mesocosm, alterations to light intensity, and changes to bacterial abundance were responsible forvariations in H2O2 concentration between the mesocosm vessels and the surrounding seawater. This study represents an interesting opportunity to observe how standard methods of experimental design (mesocosms) could potentially influence experimental outcomes in marine environments. Additionally, this study is unique in how the authors explore the effect of organisms of higher trophic levels upon H2O2 concentrations. The authors were able to provide convincing evidence supporting the importance of bacterial communities in modulating H2O2 concentrations in the ocean.

Major comments: A major conclusion of the paper is that light treatment (ambient versus artificial) has a big impact on the H2O2 concentrations in the mesocosm experiment. While this is supported by the figures, it is difficult to tell which light treatments

are used for each figure, and there is no indication in Table 1 if the mesocosms are exposed to sunlight or light bulbs.

Reply: We have made this important clarification throughout the text. Extra lines are added in Table 1 to state the exact light 'setup' for each experiment and within the text we have clarified which experiments were outdoor/indoor lighting arrangements.

Along these lines, there is essentially no discussion of the differences in light exposure, particularly the ability of UV in sunlight to generate the H2O2, and this should be mentioned in both the introduction and the discussion.

Reply: Information is added to the introduction to briefly outline the concept, "Quantum yields for H2O2 formation increase with declining wavelength and so the ultraviolet (UV) portion of natural sunlight is a major source of H2O2 in surface aquatic environments (Cooper et al., 1988, 1994). Sunlight normalized H2O2 production rates therefore peak between wavelengths of 310-340 nm (Kieber et al., 2014)." Additionally, we further add a description of the lighting different and the ability of HDPE to remove/reduce UV light in the discussion, "….considering the light arrangements for these experiments (Table 1). The Gran Canaria experiment was practically unshaded with surface seawater exposed to natural sunlight. The closed HDPE mesocosms (MesoMed, MesoPat, MesoArc) experienced natural sunlight but after attenuation through 1-2 cm of HDPE plastic. Whilst the transmission of different light wavelengths through these HDPE containers was not tested during our experiments, 1-2 cm of polyethylene should strongly attenuate the UV component of sunlight. The 20 L scale experiments (MultiMed, MultiPat, MultiArc and MicroPat) were conducted using identical synthetic lighting with lamps selected to as closely as possible replicate the wavelength distribution of natural sunlight. However, the fluorescent light distribution is still deficient, relative to sunlight, in wavelengths <400 nm, which is the main fraction of light that drives H2O2 formation in surface seawater (Kieber et al., 2014), and these containers still mitigated the limited UV exposure with a 1 mm HDPE layer which would further reduce the UV component of incoming light…"

The authors attempt to demonstrate how aspects of an experimental design (structure of vessel, setup, nutrient addition, increased stress) could affect the concentration of $H_2O_2$. While changes in $H_2O_2$ are measurable in all mesocosm experiments and are potentially attributable to a particular aspect of the experiment, the observed changes in $H_2O_2$ concentration are small with respect to total daily production of $H_2O_2$. All but one of the mesocosm experiments have $H_2O_2$ concentrations below 100nM and ranges of variation between 20-50nM. The prospect of changes in $H_2O_2$ concentration such as these recorded altering experimental outcome for microbial activity and DOC decay seems unlikely, without cited support.

Reply: These changes are certainly small and it is doubtful that the variation between different treatments within the mesocosms/multistressor experiments had measurable effects. However the side experiment in Gran Canaria did suggest a positive effect on bacteria when water was subject to a $H_2O_2$ decline equivalent to the 'gap' between natural and incubated water during some of these experiments. Nevertheless, we acknowledge that diurnal changes in $H_2O_2$ are large, and this large variation complicates any data interpretation about temporal changes in daily mean $H_2O_2$. This is now explicitly stated in the text, "A specific challenge with determining the effect(s) of $H_2O_2$ concentration on any biogeochemical processes, and vice-versa, is that the diurnal variability in $H_2O_2$ concentration is always large compared to inter-treatment differences in $H_2O_2$ concentration within individual experiments (e.g. Fig. 11). . ..."

Pg. 18 lines 24-26 – As stated here, no clear trends can be defined between $H_2O_2$ concentration and grazer abundance when considering all datasets used. Perhaps it would be beneficial to focus more intently upon the aspect of bacterial abundance and its effect upon $H_2O_2$ concentrations instead? Along with above comment, bacterial abundance is an integral part of this study's conclusions yet only 2 figures give any data on how their abundances are changing. Inclusion of cells count data for the other experiments and datasets would strengthen this major argument of the paper.

Reply: This is perhaps clear after we present the data. The logic behind a focus on zooplankton/pH/DOC was that these were gradients which were present in all experiments that could [we thought] plausibly affect equilibrium H2O2 concentrations. It wasn't clear until after looking at the data that no clear effect of zooplankton (or pH) on H2O2 was evident. We presently show bacterial productivity data for all experiments and are not sure that it is necessary to plot cell counts and productivity separately in addition to the synthesis of all data (Fig. 10). In the case of bacteria as a H2O2 sink, an additional complication is the very low H2O2 concentrations at the end of all MultiPat/Arc/Med experiments which makes it challenging to find changes in [H2O2] due to the reduced signal:noise ratio. More importantly, there is also a biological issue here (which we now mention in the text – our discussion concerning the role of bacteria (s 4.1) is expanded), because microbial organisms may adapt the strength of their oxidative defenses to ambient H2O2 concentrations i.e. cellular H2O2 defences are less active at lower H2O2 concentrations. Even for those experiments where detailed counts (total, or species level), are available, it therefore becomes difficult to make any valid argument concerning cell counts and group/species level abundances at these low H2O2 concentrations as the relationship between cell counts and H2O2 concentration would only likely be observed at higher H2O2 concentrations. "the H2O2-defence mechanism of organisms may also be sensitive to ambient H2O2 concentrations. Morris et al., (2016) suggest that microbial communities exposed to high H2O2 have elevated H2O2 defences. If the microbial communities here exhibited a dynamic response to H2O2 concentrations in terms of their extracellular H2O2 removal rates, this would dampen the correlation between bacterial abundance and H2O2 concentrations- especially at low H2O2 concentrations...."

Minor comments: The authors claim that the isolation of seawater in mesocosm vessels allows for the accumulation of H2O2. This is discussed throughout the manuscript but notably in Figure 1. on pg. 9 line 22-32 and pg. 21 line 1-11. In Figure 1, the authors claim that there is no clear trend between H2O2 and pCO2 concentration, leading them to conclude that changes in H2O2 are due to the enclosure used to house the water. Does this graph show H2O2 concentrations in unamended seawater within one of the

**BGD**

polyurethane bags used, i.e. is the baseline 400atm a control? If not, then H2O2 production cannot solely be attributed to the container used. In Figure 1 is it possible that the microbes are nutrient depleted by day 8-9, and the increase in H2O2 is due to their decline in abundance? This would also explain why the H2O2 concentration decreases around day 18 when the nutrient addition was made.

Reply: for the experiment shown in Figure 1, yes the 400 atm 'treatment' is a control in this sense i.e. atmospheric PCO2 with no additions of CO2 made (and no other additions of any kind before the nutrient spike on day 18).

Axis labels throughout manuscript are misleading. H2O2 / nM should be shown as H2O2 (nM), etc. In Figure 2 panel a, the H2O2 concentrations for ambient seawater and LG 2C treatment are difficult to discern. Consider a different representation of the data.

Reply: amended accordingly.

Pg. 20 lines 15-20 – The authors are comparing H2O2 production ranges from open ocean environments to those measured in coastal environments.

Reply: This is now explicitly stated in the text, but does not really affect our interpretation. The key point was that some diurnal ranges in mesocosms are very high (higher than expected based on diurnal ranges in the same location) whereas some diurnal ranges in mesocosms are very low (based on diurnal ranges in the same location). The offshore values are shown for comparison only to help interpret the data.

In Table 2 on pg. 20, the upper H2O2 concentrations listed for the Crete and Patagonia locations are significantly higher than any data shown in previous figures from those same locations.

Reply: These refer to 'natural' seawater outside the experiments and are included for reference only to compare to the experimental results. This further clarified both in the text and in the abstract to avoid confusion.

Pg. 21 lines 13-14 – Were individual microbial groups ever quantified? Or was this observation made from total cell counts?

Reply: For these experiments groups were quantified.

Figures 4a and 5a: are these data from the same experiment? The values for "LG 1C" look different in these figures, as one example.

Reply: No they are different datasets. 4(a) is MultiMed. 5(a) is MultiPat. We have reformatted the figure descriptions to highlight the experiment names better and avoid confusion.

---

## Author Comment (AC3) · 20 Dec 2018

Two reviewers are thanked for insightful comments on the submitted text. We respond to all of Dr Ma's comments here (some of which were posted earlier as a 'comment' rather than a 'review').

This work provides large scale mesocosm experiments to elucidate how microbial groups affect extracellular H2O2 concentrations and other related questions. It has shown that the high bacterial densities were associated with low H2O2. This

manuscript generally reads well and presents a good rationale of research. However, the study could be significantly improved with the addition of missing details on the methodology used in experiment design, as well as statistical support. The major issue is that there are so many variables in this work, which have not been fully considered regarding to the result interpretation. All these variables could play a great role in affecting the extracellular H2O2 concentration while the rationale to use these variables were not explained well and when the conclusion could not be obtained between microbial groups and H2O2 concentrations if all other variables were playing great role in it. These variables include (not limited to): zooplankton concentrations, different bacterial community, temperature, nutrient (concentrations and chemicals), light (light cycle and light intensity), DOC and pH. For example: In Glippa et al., 2018, "Vehmaa et al. [21] found that a 3 degrees rise in temperature increased the antioxidant capacity (ORAC, Oxygen Reactive Absorbance Capacity) in Acartia copepods by almost 15%, and they measured a 2-fold increase also in oxidative damage, measured as lipid peroxidation".

Reply: There are of course many variables which exert influence on extracellular H2O2 concentrations. One the main rationale for working with mesocosm experiments was that intra-experiment data is free from variation in some of these variables. Salinity/temperature/light exposure/nutrient addition are close to constant across the mesocosm units within each experiment. We have added a paragraph to explain this rationale (below). Concerning between-experiment differences, these are of course more challenging to explain because there are differences in physical/biogeochemical parameters between fieldsites. This is a main reason why we attempted to 'normalize' data to ambient H2O2 concentrations as this (and some tests on our experiment setup) provides the strongest evidence that low H2O2 across many of the experiments arises simply from the plastic containers used rather than 'natural' parameters. ". . ..our rationale for the investigation of H2O2 trends during these 20-8000 L scale mesocosm and microcosm experiments is that the experiment matrixes for each experiment permitted the changing of 1,2 or 3 key variables (DOC, zooplankton, pH) whilst maintain others (e.g. salinity, temperature, light) in a constant state across the mesocosm/microcosm

experiment. The relationships between H2O2 and other chemical/biological parameters are therefore potentially easier to investigate than in the ambient water column where mixing and the vertical/lateral trends in H2O2 concentrations must also be considered. Additionally, two of the experiment designs described herein (see Table 1) were repeated in 3 geographic locations facilitating direct comparisons between the experiment results with only limited mitigating factors concerning method changes."

Specific comments: The line numbers started over on each page. It is better to have continuous line number from the beginning to the end of the manuscript.

Reply: Changed in Revised text.

P9 L27: Is there statistics to support the "H2O2 was generally elevated"?

Reply: A line is now added in the revised text. In this particular case, the difference was so large we didn't think it necessary to detail ANOVA results, the mean/median ambient level is at least 40% lower than any treatment.

P11 L9-L10: It is hard to get the conclusion of "this trend closely matched that observed in zooplankton biomass" by only eyeballing it, especially when the 5th day of zooplankton biomass was not shown in the figure.

Reply: a reason why there is no statistical test here is because, for logistical reasons which we acknowledge are not ideal, the zooplankton biomass data and the H2O2 data are at different timepoints. There isn't a 'missing' datapoint, there is simply a lower resolution for zooplankton data in this experiment and a temporal mismatch between the two data series. One of the experimental problems, which we raise in the text already is that any inter-day temporal trend in [H2O2] made using 'spot' measurements must be done at the same time daily. Where possible (and basically wherever there are stats present in the manuscript), we timed the measurement of all parameters to be the same so that we can directly compare [H2O2] to other parameters and report [H2O2] at the same time daily. However, for some parameters, including zooplankton

during MesoMed, such a coherent timing simply wasn't possible due to the significant amount of time required to sample these parameters from the mesocosms. In these experiments, where we can only comment on the general trend, we have rephrased the text to highlight the uncertainty. The line referred to (P11 L9-10) is removed.

P12 L13: Statistics would be helpful to support "a clear difference was noted between".

Reply: t test added comparing the two groups (p <0.001) accordingly.

P13 L7-L8: Again statistics would be needed to the statement "there was a more pronounced increase".

Reply: regression/standard error details added (HG 0.31 $\pm$ 0.1, LG 1.2 $\pm$ 0.1) accordingly.

P13 L1-L13: Regarding to the statements, "In the low pH treatment (initially 7.54 $\_$ 0.09), H2O2 concentrations were significantly higher (Mann-Whitney Rank Sum test p 0.02) compared to the unmodified pH treatment (initially 8.01 $\_$ 0.02)". Only by eye-balling it, it showed the LG0.5C LpH and LG 1C LpH have higher concentration of H2O2. Is this statement based on only these two data points? Regarding to the statistics p value, it would be helpful if it is equal to, less than or greater than some certain number by indicating with corresponding symbols.

Reply: P values are now labelled < / > / = . Yes there are two very high H2O2 values in this dataset, both of which happen to be low pH/medium carbon treatments. If these values are excluded then the significance of the difference between low pH and high pH treatments disappears. Whilst there are only a limited number of datapoints in each (low/high) pH category, these two can be defined as anomalies based on 1.5 IQR if we look at the low pH and normal pH sets as groups of 8. This is now noted in the text.

P15 L8-L13: It would be great to put these discussions after (Table 1) under Discussion.

Reply: amended.

[Figure]

P16 L16-L17: Regarding to this statement, "Bacterial production showed no statistically significant (ANOVA, P 0.562) difference between low, medium and high H2O2 treatments.", there is no data to support it. Is it related with Fig. 9(c)?

Reply: No this is a separate side experiment. We had included a figure to show these data but dropped it to save space. The values (triplicate $\pm$ SD) are now provided within the text... "Bacterial production showed no statistically significant (ANOVA, p=0.562) difference between triplicate low (1.69 $\pm$ 0.28 $\mu$g C L-1 day-1), medium (1.30 $\pm$ 0.60 $\mu$g C L-1 day-1) and high (1.29 $\pm$ 0.56 $\mu$g C L-1 day-1) H2O2 treatments"

P17 L3: The author claimed there is NO significant difference while the p value is less than 0.05.

Reply: Typo corrected, should have been '> 0.05' not '< 0.05'

Figure 1: There is line to indicate the Mean H2O2. However, it is not clear on how to get this Mean.

Reply: Clarified in the figure label. ... "Data from Hopwood et al., (2018). The mean ($\pm$ SD) H2O2 from all 8 pCO2 treatments is shown"

Figure 2: Is there any interpretation on the big variation of H2O2 in ambient? Is there replicates to have error bar? Statistics would be helpful here to show the difference between HG/LG status.

Reply: We can of course speculate. The 'ambient' measurements always refer to the coastal ocean. Unlike the other fieldsites (Svalbard, Patagonia, Gran Canaria), this location (for the Mediterranean/Crete experiments) was not a sheltered fjord or harbor which likely means the H2O2 is much more variable due to changing stratification in the water column. But as we only sampled surface water at intervals during the experiment we can't really quantify this or do anything other than speculate about the underlying causes.

The discussion of the zooplankton trend is now not explicitly linked to H2O2 (see comment above). Noting the different timing of the measurements during this specific experiment it is not possible to produce meaningful statistics.

There are replicate measurements for all ambient water measurements, which produce a very small error bar (1-5%). However, given the short-term changes to H2O2 that can occur in a dynamic water column even on very short (minutes) timescales (as demonstrated in our high resolution diurnal time series) we thought that plotting error bars based on analytical error for spot measurements would be misleading as it is not inclusive of the changes to [H2O2] that occur in natural waters over a time period equivalent to the sample collection/measurement time of 10-20 minutes.

Figure 7: It would be great to show diurnal cycling of H2O2 in two continuous days.

Reply: It would, but when the apparatus is set up to produce continuous data like this an analyst has to check on the instruments very regularly. It simply wasn't possible here to have them operating for more than 24 hours! We may try a different instrument/sensor configuration to achieve this in the future with slightly lower resolution and an auto-clean cycle.

---

## Referee Report (RR1)

This study attempts to elucidate issues (both natural and induced by experimental design) that affect production of hydrogen peroxide in marine environments. The implications of this study may be very important to marine ROS researchers around the world, and as such the science presented is important.

The experiments in this study cover a wide variety of factors and each incubation had a slightly different set-up. While this allowed the authors to address many aspects of experimental design, it is difficult for the reader to keep track of everything that was done. Accordingly, it is very important for the authors to present an organized overview of all experiments. Table 2 is a good start on doing this but additional work should be done before publication.

As with any study that covers such a broad array of factors, care needs to be taken when trying to draw specific conclusions. In general, the authors did a good job of indicating where the impact of a particular factor was uncertain. The primary concern I have is with the impact of temporal variations in $H_2O_2$. The authors address diel cycling of $H_2O_2$ concentrations briefly in both the introduction and discussion sections, and even show a graph of diurnal variation in $[H_2O_2]$ in figure 7. However, most of the temporal $H_2O_2$ graphs show only one data point per day, and there is no mention of what time of day measurements were taken. If the $H_2O_2$ was sampled at irregular times of day, that in and of itself could account for all of the fluctuations seen in the temporal graphs—which would negate any conclusions drawn regarding other factors' impact on $H_2O_2$ production. Publication should not occur until this issue is addressed in a more overt fashion.

| | |
|---|---|
| Line 26 | typo: should be "mis-match" |
| Line 46 | The authors mention production as being primarily photochemical. However, there is substantial evidence of dark (most likely biological) production of ROS in the marine environment (see, for example, Vermilyea et. al., 2010, *Limnol. Oceanog.*; Hansard et al., 2010, *Deep Sea Res. I*). The authors bring this up in the discussion but there is no mention of it in the introduction. This seems like a substantial oversight. |
| Line 71 | The authors acknowledge that some community members do not have the ability to remove $H_2O_2$. It has also been shown that community members who do have the ability to remove $H_2O_2$ may not actively express this ability. (Morris et al., 2016, *J. Plankton Res.*) This paper is cited in the discussion, but it should also be addressed in the introduction. |
| Line 80 | Later in the paper, the authors discuss light as a significant factor in $H_2O_2$ production, yet it is not mentioned in the list of factors tested. |
| Line 105 (Table 1) | I did not find this table helpful. For example, the items in the table are organized differently than in the text, which made it confusing for the reader to use as an aid in keeping the different incubations straight. Much of the data can be found in Table 2, which is far better organized. I would expand Table 2 to incorporate additional data from Table 1. |
| Line 110 (Table 2) | Add info on glucose addition to the macronutrient data (or in a different row) so that all pertinent data on incubations is found in this table. Additionally, information on the ancillary experiments described in lines 159-184 should be added to Table 2. |
| Line 195 | There is no mention of when (i.e. what time of day) the $H_2O_2$ samples were taken. |

Given the importance of diel cycles in $H_2O_2$ concentrations—the authors make note of this in line 544—this is a major oversight. If data points were taken at inconsistent times, that should be noted here as it would allow readers to better interpret the data.

Line 280
(Figure 3)

There is no mention of what the error bars represent.

Line 380
(Figure 9)

The authors discuss both $H_2O_2$ and cell counts for both Mediterranean and Grand Canaria locations. However, only cell counts are shown for GC. I think it would be best to redo this figure as a set of four graphs: two graphs (one for each location) for $H_2O_2$ concentration on the first row, and two graphs for bacterial abundance in the second row. This would more strongly contrast the difference between incubations with added nutrients and no added nutrients.

---

## Author Response (AR2)

[revised manuscript text omitted]

705   Reply to reviewer comments:

This study attempts to elucidate issues (both natural and induced by experimental design) that affect production of hydrogen peroxide in marine environments. The implications of this study may be very important to marine ROS researchers around the world, and as such the science presented is
710   important.
The experiments in this study cover a wide variety of factors and each incubation had a slightly different set-up. While this allowed the authors to address many aspects of experimental design, it is difficult for the reader to keep track of everything that was done. Accordingly, it is very important for the authors to present an organized overview of all experiments. Table 2 is a good start on doing
715   this but additional work should be done before publication.

**R: As requested below, Tables 1 and 2 are merged with additional entries made into Table 2 to present a single, complete record of all experimental work.**

720   As with any study that covers such a broad array of factors, care needs to be taken when trying to draw specific conclusions. In general, the authors did a good job of indicating where the impact of a particular factor was uncertain. The primary concern I have is with the impact of temporal variations in $H_2O_2$. The authors address diel cycling of $H_2O_2$ concentrations briefly in both the introduction and discussion sections, and even show a graph of diurnal variation in $[H_2O_2]$ in figure 7. However,
725   most of the temporal $H_2O_2$ graphs show only one data point per day, and there is no mention of what time of day measurements were taken. If the $H_2O_2$ was sampled at irregular times of day, that in and of itself could account for all of the fluctuations seen in the temporal graphs—which would negate any conclusions drawn regarding other factors' impact on $H_2O_2$ production. Publication should not occur until this issue is addressed in a more overt fashion.
730

**R: We agree entirely with this comment; variation in the time of day of sampling would render any daily-resolution time series challenging to interpret. The submitted manuscript did however state that 'H2O2 concentration varies on diurnal timescales and thus during each experiment where a time series of H2O2 concentration was measured, sample collection and analysis occurred at the**
735   **same time daily' (original line pg7 Line 14). For extra clarity on this issue we have added an additional sentence detailing the exact time of day for each experiment where time series are constructed: 'For MesoMed sampling occurred at 14:40, for Gran Canaria at 11:00 (local times). Sample times were selected to be intermediate with respect to the diurnal cycle (with peak $H_2O_2$ expected mid-afternoon, and the lowest $H_2O_2$ expected overnight).'**
740

Line 26 typo: should be "mis-match"
**R:corrected.**
Line 46 The authors mention production as being primarily photochemical. However, there

is substantial evidence of dark (most likely biological) production of ROS in the
marine environment (see, for example, Vermilyea et. al., 2010, *Limnol. Oceanog.*;
Hansard et al., 2010, *Deep Sea Res. I*). The authors bring this up in the discussion but
there is no mention of it in the introduction. This seems like a substantial oversight.
**R: Yes, this was an oversight. An additional few lines are added here for completeness. 'In addition to photochemical generation of ROS in the photic zone, there is ample evidence of dark formation processes for H2O2 in both surface and sub-surface waters.'**

Line 71 The authors acknowledge that some community members do not have the ability to
remove $H_2O_2$. It has also been shown that community members who do have the
ability to remove $H_2O_2$ may not actively express this ability. (Morris et al., 2016, *J. Plankton Res.*) This paper is cited in the discussion, but it should also be addressed in
the introduction.
**R: Agreed, yes this citation is now added here. 'enzymes are widely produced by many, but not all, marine microbes to lower extracellular H2O2 concentrations. Furthermore, some community members possessing the ability to remove H2O2 may not actively express this ability constantly, with H2O2 defenses thought to be subject to diurnal regulation.'**

Line 80 Later in the paper, the authors discuss light as a significant factor in $H_2O_2$
production, yet it is not mentioned in the list of factors tested.
**R: Rephrased 'changes in DOC, pH, ambient light conditions and grazing pressure'**

Line 105 I did not find this table helpful. For example, the items in the table are organized
(Table 1) differently than in the text, which made it confusing for the reader to use as an aid in
keeping the different incubations straight. Much of the data can be found in Table 2,
which is far better organized. I would expand Table 2 to incorporate additional data
from Table 1.
Line 110 Add info on glucose addition to the macronutrient data (or in a different row) so
(Table 2) that all pertinent data on incubations is found in this table. Additionally, information
on the ancillary experiments described in lines 159-184 should be added to Table 2.
**R: Tables 1 and 2 are merged with additional lines added to Table 2 as suggested. We attempted adding the smaller experiments as well, but this became messy and so have left these in the main text.**

Line 195 There is no mention of when (i.e. what time of day) the $H_2O_2$ samples were taken.
Given the importance of diel cycles in $H_2O_2$ concentrations—the authors make note
of this in line 544—this is a major oversight. If data points were taken at inconsistent
times, that should be noted here as it would allow readers to better interpret the data.
**R: A few sentences did explicitly confirm that timer series were conducted at the same time of day, in the revised text we additionally at the timing of sample collection to clarify this further: 'For MesoMed sampling occurred at 14:40, for Gran Canaria at 11:00 (local times). Sample times were selected to be intermediate with respect to the diurnal cycle (with peak H$_2$O$_2$ expected mid-**

**afternoon, and the lowest $H_2O_2$ expected overnight).'. The time of day was standardized for all time series for this exact reason.**

Line 280 There is no mention of what the error bars represent. (Figure 3)

790 **R: Clarified for each figure (in this case, standard deviation of triplicate measurements)**

Line 380 The authors discuss both $H_2O_2$ and cell counts for both Mediterranean and Grand (Figure 9) Canaria locations. However, only cell counts are shown for GC. I think it would be best to redo this figure as a set of four graphs: two graphs (one for each location) for

795 $H_2O_2$ concentration on the first row, and two graphs for bacterial abundance in the second row. This would more strongly contrast the difference between incubations with added nutrients and no added nutrients.

**R: Yes this is straightforward, now amended to four graphs with H2O2 and counts for each location.**

800

---

## Author Response (AR3)

The reviewer is thanked for their time and detailed comments on the text.

*On page 3, line 20: In these mesocosm experimental setup, HDPE containers were used. H2O2 is mainly produced through UV oxidation of dissolved organic material. How much UV radiation can be transmitted through the HDPE wall of these containers?*

**Given the thickness of any plastic 'tanks' UV should have been strongly attenuated with little transmission through the HDPE wall. This is now explicitly stated in the text. Line added: 'In all cases, these HDPE containers likely strongly attenuated natural UV radiation, compared to ambient waters, which is expected to negatively affect photochemical formation of H2O2 (Cooper et al., 1988, 1994).'**

*On page 6, line 30: When the artificial light source is used in the experiment. How much UV light is provided by these artificial light sources?*

**We did not quantify this explicitly relative to ambient light, but based on the lamp's spractral distribution, they are deficient in UV, especially compared to sunlight at low latitudes. We discuss this briefly later, but also now flag here as well, 'Whilst the light condition for these experiments was selected to approximate the light intensity of ambient near-surface waters, the synthetic lighting is deficient in UV relative to ambient sunlight-especially at low latitudes.'**

*On page 7, line 20: I believe residue catalase on containers can affect H2O2 sampling and determinations, what cautious steps were taken to avoid that?*

**This is possible, but we did not see any evidence of analytical issues herein (which would have been evident in our standard additions of H2O2 during calibration). The plasticware used to prepare catalase was discarded, and our measurements (in catalase treated incubations) were made >12 h after any manipulation with catalase. Any added catalase was likely largely de-activated between addition and H2O2 measurements (catalase additions occurred at sunset, measurements of H2O2, as reported, were made the following day). An extra line of detail is added. 'Plasticware used to handle catalase was discarded. No adverse effects of measuring H2O2 in catalase-manipulated solutions were found. As H2O2 measurements were made >12 h after catalase addition, this may reflect catalase de-activation under the incubated conditions.'**

*On page 8, line 11: Fe(II) can cause severe interferences in luninol based chemiluminescence determination of H2O2. For discrete samples, a short delay from sampling to determination may be enough time for Fe(II) to decay away, what measures were taken to avoid Fe(II) interferences in continuous sampling and determinations?*

**Yes this is certainly the case. In a companion text we describe Fe(II) concentrations for the same mesocosms discussed herein which included measurements of Fe(II) in ambient waters. For continuous measurements of H2O2 'in situ', the apparatus had a loop to induce a delay of 1-2 minutes in the dark. Any short-lived radial species would thereby be reduced in concentration prior to measurement of H2O2. Fe(II) was below detection (<0.2 nM) in all the Mediterranean experiments discussed herein. In Patagonia/Svalbard/Gran Canaria, Fe(II) was measurable in situ and a residual Fe(II) signal was present in the dark-especially in Patagonia. Without any Fe(II) decay over 1-2 minutes, the maximum possible over-estimate of H2O2 in-situ would be <10%. We explicitly flag this uncertainty, it is challenging to be more accurate as Fe(II) and H2O2 are difficult to determine at the exact same time. 'Fe(II) and photochemically generated radical species can interfere with the luminol based chemiluminescence used to determine $H_2O_2$. In batch measurements, the >15 minute time delay between sample collection analysis (during which time**

the same in not exposed to light) is likely sufficient to minimize interference from these species which have much shorter half-lives and lower concentrations than $H_2O_2$. For in situ, continuous measurements, a sample loop was intentionally introduced such that seawater was displaced from ambient light for 2 minutes prior to analysis. Some residual Fe(II) may have therefore remained leading to over-estimation of $H_2O_2$ concentrations. As Fe(II) concentrations were quantified in all of the experiments and ambient waters described herein, we can determine that the maximum possible over-estimate of $H_2O_2$ concentrations for ambient waters during continuous analysis was <10%.'

*On page 11, line 21: the artificial light source may not have provided enough UV radiation to match that of sunlight.*

**Ammended 'with natural lighting and especially higher UV light exposure'**

*On page 14, line 9: "diel" should be used in this situation instead of "diurnal".*

**Ammended.**

*On page 21, line 6: UV transparency of HDPE and polyurethane may explain the differences here.*

[revised manuscript text omitted]